# Cigarette smoke aggravates asthma by inducing memory-like type 3 innate lymphoid cells

Jongho Ham [1,2,10], Jihyun Kim [1,3,10], Kyoung-Hee Sohn [4], In-Won Park [5], Byoung-Whui Choi[5,6], Doo Hyun Chung[2,7,8], Sang-Heon Cho[3,9], Hye Ryun Kang [3,9], Jae-Woo Jung [5✉] & Hye Young Kim [1,2,3✉]

Although cigarette smoking is known to exacerbate asthma, only a few clinical asthma studies have been conducted involving smokers. Here we show, by comparing paired sputum and blood samples from smoking and non-smoking patients with asthma, that smoking associates with significantly higher frequencies of pro-inflammatory, natural-cytotoxicity-receptor-non-expressing type 3 innate lymphoid cells (ILC3) in the sputum and memory-like, CD45RO-expressing ILC3s in the blood. These ILC3 frequencies positively correlate with circulating neutrophil counts and M1 alveolar macrophage frequencies, which are known to increase in uncontrolled severe asthma, yet do not correlate with circulating eosinophil frequencies that characterize allergic asthma. In vitro exposure of ILCs to cigarette smoke extract induces expression of the memory marker CD45RO in ILC3s. Cigarette smoke extract also impairs the barrier function of airway epithelial cells and increases their production of IL-1β, which is a known activating factor for ILC3s. Thus, our study suggests that cigarette smoking increases local and circulating frequencies of activated ILC3 cells, plays a role in their activation, thereby aggravating non-allergic inflammation and the severity of asthma.

[1] Laboratory of Mucosal Immunology, Department of Biomedical Sciences, Seoul National University College of Medicine, Seoul, Republic of Korea. [2] Department of Biomedical Sciences, BK21 Plus Biomedical Science Project, Seoul National University College of Medicine, Seoul, Republic of Korea. [3] Institute of Allergy and Clinical Immunology, Seoul National University Medical Research Center, Seoul, South Korea. [4] Department of Internal Medicine, Kyung Hee University Medical Center, Seoul, Republic of Korea. [5] Department of Internal Medicine, Chung-Ang University College of Medicine, Seoul, South Korea. [6] Department of Internal Medicine, Chung-Ang University H.C.S. Hyundae I Hospital, Namyangju, South Korea. [7] Department of Pathology, Seoul National University College of Medicine, Seoul, South Korea. [8] Laboratory of Immune Regulation, Department of Biomedical Sciences, Seoul National University College of Medicine, Seoul, South Korea. [9] Department of Internal Medicine, Seoul National University Hospital, Seoul, South Korea. [10]These authors contributed equally: Jongho Ham, Jihyun Kim. ✉email: jwjung@cau.ac.kr; hykim11@snu.ac.kr

A sthma is a heterogeneous, chronic inflammatory disease that associates with bronchial obstruction and excessive mucus production. It was initially thought to be a solely allergic disease that is mediated by type 2 immune responses, including the production of interleukin (IL)−4, IL-5, IL-9, and IL-13. These cytokines typically associate with immunoglobulin E production, eosinophilia, mast cell activation, and mucus secretion by airway epithelial cells[1]. However, it has been shown more recently that 10–33% of people with asthma have nonallergic asthma (i.e., asthma in which allergic sensitization cannot be demonstrated)[2]. Unlike allergic asthma, nonallergic asthma is triggered by ozone exposure, obesity, and air pollutants; it is also typically more severe than allergic asthma[3]. However, the causes of nonallergic asthma remain poorly understood and there are few studies on asthma with low or non-type 2 helper T cells (Th2)-mediated inflammation. As a result, although several therapies are currently under clinical development for asthma characterized by Th2 immune responses, the same is not true for non-allergic asthma.

Cigarette smoking has adverse health effects[4]. In particular, there is a close correlation between smoking and asthma, and many clinical studies show that cigarette smoke significantly reduces lung function and increases the risk of developing asthma[5]. Smoking also enhances the risk of pathogen invasion by destroying the epithelial barrier and impairing airway tight junctions[6]. In addition, cigarette smoke has known immuno-modulatory properties throughout the body[7], including increasing alveolar macrophage numbers in the airway and elevating neutrophil recruitment to the lung[8]. However, many questions remain, including how cigarette smoke changes airway immunity in asthma and whether these changes increase asthma severity.

Innate lymphoid cells (ILCs) are recently discovered lymphoid cells that reside and self-renew in mucosal tissues[9]. There are three types of innate lymphoid cells, namely, ILC1, ILC2, and ILC3. They lack antigen-specific receptors and are instead rapidly activated by innate cytokines that are secreted in the site of inflammation. Once activated, they control immune responses by secreting specific effector cytokines: ILC1s, ILC2s, and ILC3s, respectively secrete interferon (IFN)-γ, IL-5, and IL-13, and IL-17A and IL-22[10]. In the present study, we focused on the effect of cigarette smoking on ILCs for several reasons. First, ILCs are key coordinators of immunological homeostasis and immunity in the respiratory tract[11,12]. Second, many studies have highlighted the importance of ILC2s in the development of allergic asthma while ILC1s and ILC3s appear to contribute to the pathogenesis of non-allergic asthma[11,12]. Third, we reported previously that ILCs not only modulate macrophage polarization in the airways of asthma patients, different ILC populations are also expanded in different types of asthma[13]. Finally, despite the reported roles of distinct ILC subsets in various diseases[14], little is known about airway ILCs in smokers, let alone about these cells in smokers with asthma, on whom there are no published data.

Here, we examined whether cigarette smoke alters airway immune cells, especially innate lymphoid cells, and how this affects lung function in patients with asthma. We showed that smokers with asthma have higher frequencies of induced sputum ILC3s than non-smokers with asthma; they also have higher frequencies of CD45RO-expressing memory-like ILC3s in their peripheral blood. In addition, these increased ILC3 frequencies correlated with the smoking amount, disease severity, and M1 macrophage and circulating neutrophil numbers, but not with circulating eosinophil numbers. Moreover, we showed that cigarette smoke impairs the integrity of airway epithelial cells and causes them to express IL-1β, which is critical for the proliferation and activation of ILC3s[10]. These findings suggest that cigarette smoke induces activation of ILC3s, which are involved in the

exacerbation of asthma. Therefore, ILC3s might participate in the pathogenesis of non-allergic asthma by sensing the environmental changes caused by smoking.

## Results

### Cigarette smoking in asthma associates with higher ILC3 frequencies in induced sputum.
To investigate the effect of cigarette smoking on ILCs in patients with asthma, we recruited 58 current or previous smokers with asthma, 33 non-smokers with asthma, 11 healthy smokers, and 13 healthy non-smokers (Table 1). Compared to the non-smokers with asthma, the smokers with asthma had significantly impaired lung function, as shown by (i) lower forced expiration volume of 1s% (FEV$_1$%), a ratio of FEV$_1$/FVC, and asthma control test (ACT) scores, and (ii) higher asthma control questionnaire (ACQ) scores. As indicated by the PC$_{20}$ value, the two groups did not differ in airway hyperresponsiveness (Table 1).

We first analyzed the ILC subsets in induced sputum. Total ILCs were defined as CD45$^+$Lineage$^-$IL-7R$^+$ lymphocytes, after which the ILC1s (ST-2$^-$C-kit$^-$cells), ILC2s (ST-2$^+$ cells), and ILC3s (ST-2$^-$C-kit$^+$ cells) were identified (Fig. 1a). Smoking in both asthma and healthy groups tended to associate with higher frequencies of total airway ILCs but these differences did not achieve statistical significance (Fig. 1b). Smoking in the healthy group was associated with higher ILC1 frequencies (Supplementary Fig. 1a), but this was not observed for the asthma group (Fig. 1c). The four groups had similar ILC2 frequencies (Fig. 1d and Supplementary Fig. 1a). By contrast, smoking associated with significantly greater sputum ILC3 frequencies in the asthma group (Fig. 1e) but not in the healthy group (Supplementary Fig. 1a).

ILC3s can be further subdivided according to their expression of natural cytotoxicity receptors (NCRs): NCR$^+$ILC3s secrete IL-22 to maintain epithelial barrier integrity, whereas NCR$^-$ILC3s mainly produce IL-17A and IL-17F, which are known to induce pathogenic inflammation[15]. We observed that smoking in asthma associated with a greater frequency of NCR negative sputum ILC3s (Fig. 1f). By contrast, smoking did not have this effect on the healthy controls (Supplementary Fig. 1b). These data suggest that cigarette smoking may provoke harmful changes in the ILC3 subset in the asthmatic airways.

To exclude the possibility that the smoking-related changes in ILC subset frequencies were merely due to the severity of asthma, we selected the 32 non-smokers and 34 smokers in the asthma group who had normal lung function (defined as FEV$_1$ > 80%). Again, smoking associated with higher total ILC and ILC3 frequencies in the asthmatic airway (Supplementary Fig. 2a, b).

### Smoking in asthma associates with CD45RO-expressing ILC3s in the peripheral blood.
The effect of smoking on the ILCs in the peripheral blood mononuclear cells (PBMCs) from the four groups was analyzed in the same way as for the sputum ILCs (Fig. 2a). Smoking associated with a modest (statistically insignificant) increase in total ILC frequencies in the PBMCs from asthma patients but significantly lower total ILC frequencies in the healthy group (Fig. 2b). ILC subset analyses then showed that smoking slightly increased ILC1 and ILC3 frequencies and had no effect on ILC2s in the asthma group (Fig. 2c) but significantly decreased ILC1 and ILC2 frequencies and had no effect on the ILC3s in the healthy group (Supplementary Fig. 3a). Thus, at first glance, smoking in asthma did not appear to have a marked effect on the ILC3 frequencies in the peripheral blood, unlike in the sputum.

Nonetheless, we decided to explore the PBMCs further. Since smoking significantly increased NCR$^-$ILC3s in the airway of

**Table 1 Characteristics of the study subjects.**

| | Healthy Control | | Asthma | | |
| | Non-smoker | Smoker | Non-smoker | Smoker | P value |
|---|---|---|---|---|---|
| n | 13 | 11 | 33 | 58 | |
| Age (Year) | 55.1 ± 5.7 | 59.0 ± 13.3 | 52.1 ± 14.0 | 55.1 ± 11.4 | |
| Sex (M/F) | 6/7 | 8/3 | 6/27 | 49/9 | |
| Body mass index (kg/m$^2$) | 23.0 ± 2.6 | 26.3 ± 2.7 | 23.6 ± 3.1 | 25.8 ± 4.9 | |
| Allergic rhinitis, n (%) | 0 (0) | 0 (0) | 21 (63.6) | 32 (55.2) | |
| Atopic dermatitis, n (%) | 0 (0) | 0 (0) | 3 (9.1) | 6 (10.4) | |
| FVC (mL) | 3465.0 ± 837.0 | 4002.0 ± 732.4 | 3180.0 ± 1014.0 | 3646.0 ± 878.3 | 0.1037 |
| FVC (%) | 107.3 ± 12.6 | 100.7 ± 19.1 | 103.5 ± 13.8 | 88.2 ± 15.1 | <0.0001 |
| FEV1 (mL) | 2749.0 ± 639.6 | 3177.0 ± 595.3 | 2542.0 ± 657.5 | 2444.0 ± 786.6 | 0.0747 |
| FEV1 (%) | 103.5 ± 12.6 | 104.8 ± 14.1 | 95.6 ± 10.5 | 76.1 ± 16.7 | <0.0001 |
| FEV$_1$/FVC (%) | 79.8 ± 6.4 | 79.6 ± 7.7 | 77.0 ± 5.4 | 65.9 ± 11.2 | <0.0001 |
| ACQ | 0 | 0.1 ± 0.1 | 5.8 ± 4.4 | 7.0 ± 5.3 | 0.0271 |
| ACT | 25 | 25 | 21.8 ± 3.1 | 20.2 ± 3.4 | 0.0061 |
| PC$_{20}$ (mg/ml) | n.d | n.d | 10.9 ± 9.0 | 5.5 ± 6.2 | 0.1364 |
| OCS, n (%) | n.d | n.d | 5 (15.2) | 13 (22.4) | |
| Hemoglobin (g/dL) | n.d | n.d | 13.3 ± 2.4 | 15.0 ± 1.7 | 0.0003 |
| WBC (/μL) | n.d | n.d | 7055.0 ± 2830.0 | 7948.0 ± 3650.0 | 0.2978 |
| Lymphocytes (/μL) | n.d | n.d | 1958.0 ± 740.0 | 2317.0 ± 1310.0 | 0.2548 |
| Monocytes (/μL) | n.d | n.d | 428.3 ± 153.1 | 576.8 ± 365.5 | 0.1301 |
| Eosinophils (/μL) | n.d | n.d | 317.3 ± 349.3 | 318.6 ± 314.5 | 0.6822 |
| Neutrophils (/μL) | n.d | n.d | 4174.0 ± 2510.0 | 4655.0 ± 3514.0 | 0.6754 |
| Basophils (/μL) | n.d | n.d | 39.7 ± 23.4 | 39.0 ± 22.2 | 0.6179 |

The data are presented as mean ± standard deviation of n (%).

ACT asthma control test, ACQ asthma control questionnaire, FEV1 Forced expiratory volume in one second, FVC forced vital capacity, n.d. not determined, OCS oral corticosteroid, PC20 provocative concentration of methacholine required to decrease FEV1 by 20%, WBC white blood cells.

*P values were determined by comparing the non-smoker asthma patients with the smoking asthma patients by Mann-Whitney U test. The healthy non-smoker and smoker groups did not differ significantly in lung function indices (P values not shown).

asthma patients (Fig. 1f) but circulating ILC3s do not express NCR[16], we asked whether the PBMCs of the smoking asthma patients contained higher frequencies of an airway disease-associated ILC3 subset, namely, CD45RO-expressing ILC3s. These cells are elevated in the lung and tonsils of patients with chronic obstructive pulmonary disease (COPD)[17]. Moreover, CD45RO-expressing blood ILC2s are elevated in steroid-resistant asthmatic patients[18]. While it is not yet clear what CD45RO expression signifies in terms of ILC functions, it is well-known that in T cells (the adaptive immunity equivalents of ILCs), CD45RO is a marker of activated/memory T cells while CD45RA is expressed on naïve T cells[19]. Indeed, when we assessed the CD45RA and CD45RO expression of the blood ILCs (Fig. 2d), we observed that smoking in the asthma group, but not the healthy group, associated with a modest increase of CD45RO$^+$ILC3s in the PBMCs (Fig. 2e and Supplementary Fig. 4a). By contrast, the four groups did not differ significantly in terms of circulating CD45RA$^+$ILC3s (data not shown).

Retinoic acid induces RORγt expression in NCR$^-$CD4$^-$ILC3 precursors, and RORγt differentiates them into NCR$^-$CD4$^+$ILC3s, in mice[20]. Assuming that the CD4-expressing NCR$^-$ILC3s would be in a more differentiated form, we further confirmed CD4 expression. We noted that the CD45RO$^+$ILCs could be further divided into CD4$^+$ and CD4$^-$ subsets (Fig. 2f). Analysis of these CD4$^+$ expressing subsets showed that smoking in the asthma group, but not the healthy group, associated with significantly increased blood frequencies of CD4$^+$CD45RO$^+$ILC3s (Fig. 2g and Supplementary Fig. 4b). Thus, smoking in asthma specifically increased the frequencies of CD4$^+$CD45RO$^+$ILC3s in the blood. Moreover, the blood CD4$^+$CD45RO$^+$ILCs were significantly more likely to express IL-17A, a major cytokine released from ILC3s, than the CD4$^-$CD45RO$^+$ILCs or CD45RA$^+$ILCs (Fig. 2h), which suggests that they are more active than the other subsets. Similar analyses with the asthma patients with normal lung function showed again

that although smoking did not significantly alter the blood frequencies of total ILCs or the three ILC subsets (Supplementary Fig. 5a, b), it did associate with significantly higher circulating CD4$^+$CD45RO$^+$ILC3 frequencies (Supplementary Fig. 5c).

It should be noted that smoking in the asthma group was also associated with slightly higher circulating ILC1 (Fig. 2c) and CD45RO$^+$ILC1 (Fig. 2e) frequencies; however, it did not associate with higher CD4$^+$CD45RO$^+$ILC1 frequencies (Fig. 2g). By contrast, smoking in the healthy group associated with significantly fewer ILC1s, CD45RO$^+$ILC1s, and CD4$^+$CD45RO$^+$ILC1s (Supplementary Figs. 3a and 4a, b). Thus, unlike the equivalent subset in ILC3s, CD4$^+$CD45RO$^+$ILC1s are not specifically expanded in the blood of smoking patients with asthma. This suggests that the expansion of ILC1s in the blood of smoking patients with asthma may be secondary to decreased pulmonary function rather than playing a pathogenic role.

We also examined the CD4$^+$ T cells in the peripheral blood of the four groups. While smoking associated with significantly reduced CD4$^+$ T cell frequencies in the healthy group, this was not observed in the asthma patients. The frequencies of CD4$^+$ T cells that expressed IFNγ, IL-5, or IL-17A were also similar between smoking and non-smoking asthmatics (Supplementary Fig. 6a–c). Moreover, smokers in the asthma group did not have higher frequencies of memory CD45RO$^+$ T cells than non-smokers (Supplementary Fig. 6d). Thus, memory T cells do not appear to be augmented by smoking in asthma.

**In vitro cigarette smoke exposure induces memory-like ILC3s in PBMCs from asthma patients but not healthy individuals.** Since cigarette smoke appeared to augment sputum ILC3s and circulating CD4$^+$CD45RO$^+$ILC3s in patients with asthma (but not healthy individuals), we assessed whether the smoking amount, as expressed by Pack/Year (PY), correlated with the

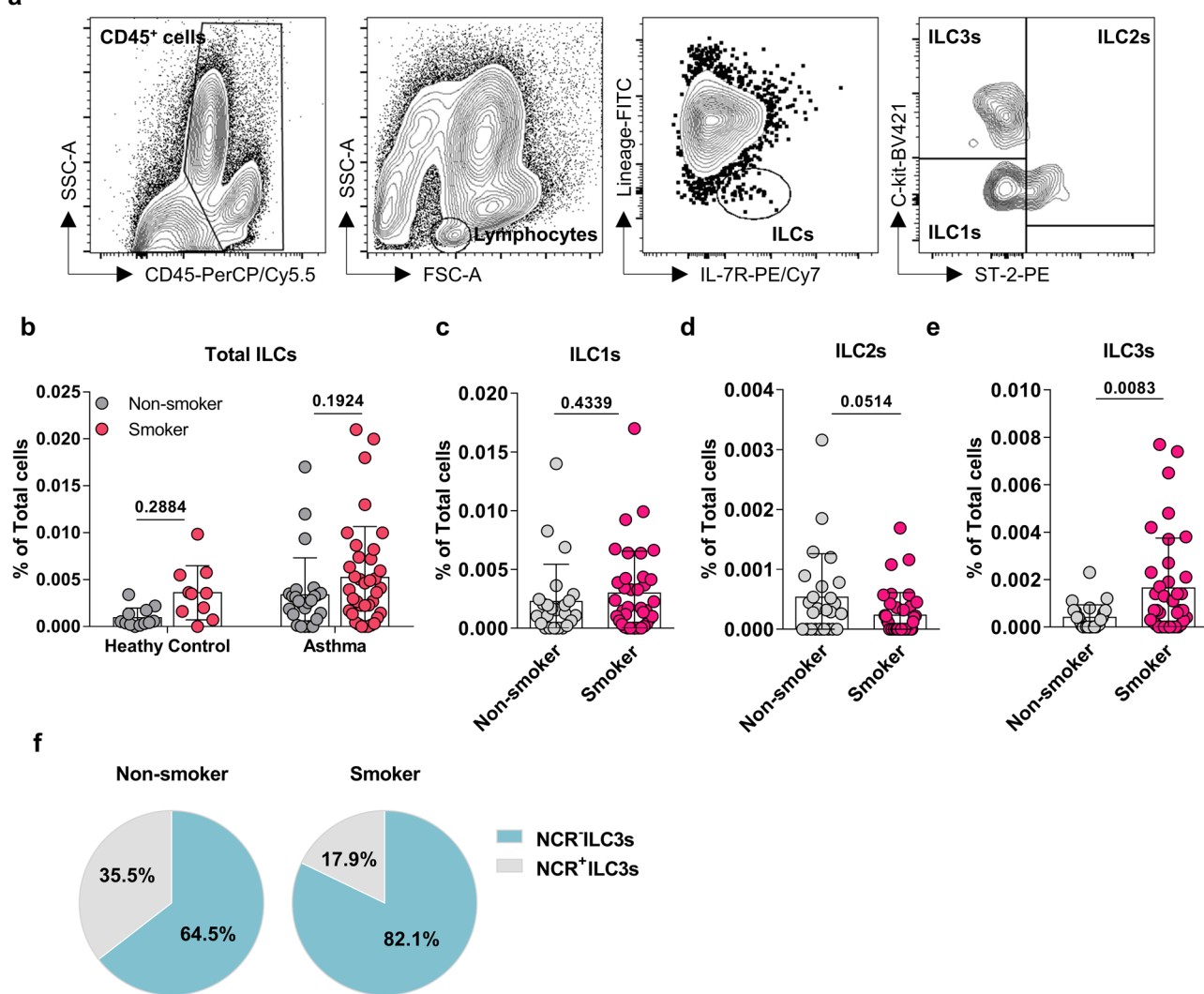

**Fig. 1 Compared to non-smoking asthma patients, asthma patients who smoke have higher ILC3 frequencies in induced sputum. a** Gating strategy for the three subsets of innate lymphoid cells (ILCs) in induced sputum. Total ILCs were gated as Lineage (CD3ε, CD11c, CD11b, CD14, CD19, CD49b, and FcεRIα)-negative and IL-7R-positive cells. ST-2 and C-kit expression was used to distinguish ILC1s (ST-2−C-kit−), ILC2s (ST-2+), and ILC3s (ST-2−C-kit+). **b** Comparison of non-smoking and smoking asthma patients and non-smoking and smoking healthy individuals in terms of total ILC frequencies in induced sputum. Comparison of non-smoking and smoking asthma patients in terms of sputum ILC1 (**c**), ILC2 (**d**), and ILC3 (**e**) frequencies. **f** The proportion of NCR− (NKp44−) and NCR+ (NKp44+) ILC3s in the induced sputum of non-smoking and smoking asthma patients. Each dot represents individual subject. Sample size of non-smoking healthy controls, $n = 12$; smoking healthy controls, $n = 10$; non-smoking asthma patients, $n = 25$; smoking asthma patients, $n = 37$ for Fig. 1b, non-smokers, $n = 25$; smokers, $n = 37$ for Fig. 1c, e, non-smokers, $n = 25$; smokers, $n = 38$ for Fig. 1d. The non-smokers and smokers in the asthma patients or healthy individuals were compared by two-way ANOVA (Fig. 1b) or two-tailed Mann-Whitney U test (Fig. 1c–e). All data are presented as mean ± standard deviation. $p < 0.05$ is considered as significant.

frequencies of these ILC3 populations in asthma patients. Indeed, both populations in the asthma group correlated positively and significantly with PY (Fig. 3a, b).

Since CD45RO is a key marker for primed memory T cells[19] and we observed that the circulating CD4+CD45RO+ILCs from asthma patients were more likely to express IL-17A than the corresponding CD4−CD45RO+ILCs or CD45RA+ILCs (Fig. 2h), we speculated that when asthma patients (but not healthy individuals) are exposed to cigarette smoke, the biological properties of ILC3s may change, including acquiring an activated memory-like phenotype (CD4+CD45RO+). To test whether the CD4+CD45RO+ circulating ILC3s are maintained after quitting smoking (i.e., whether they display a memory function), we divided the asthma patients according to their current smoking status (non-smokers, former smokers, and current smokers).

Indeed, both former and current smokers had significantly higher frequencies of CD4+CD45RO+ILC3s in the blood than non-smoker asthma patients (Fig. 3c). These data suggested that smoking induces the CD4+CD45RO+ILC3 subset that persists even after quitting smoking like memory immune cells (memory-like ILC3s).

Next, we asked whether in vitro culture of PBMCs from the non-smoking healthy and asthma groups with 0.1% cigarette smoke extract (CSE) directly altered the phenotype of their ILC populations (Fig. 3d). Treatment with CSE did not alter the CD45RA or CD45RO expression of the circulating ILC1s from any of the groups (Fig. 3e, f). By contrast, CSE treatment significantly increased the expression of CD45RO by the ILC3s in asthma patients but this was not observed in the healthy individuals (Fig. 3g). CSE treatment did not alter the CD45RA expression of the ILC3s of any group

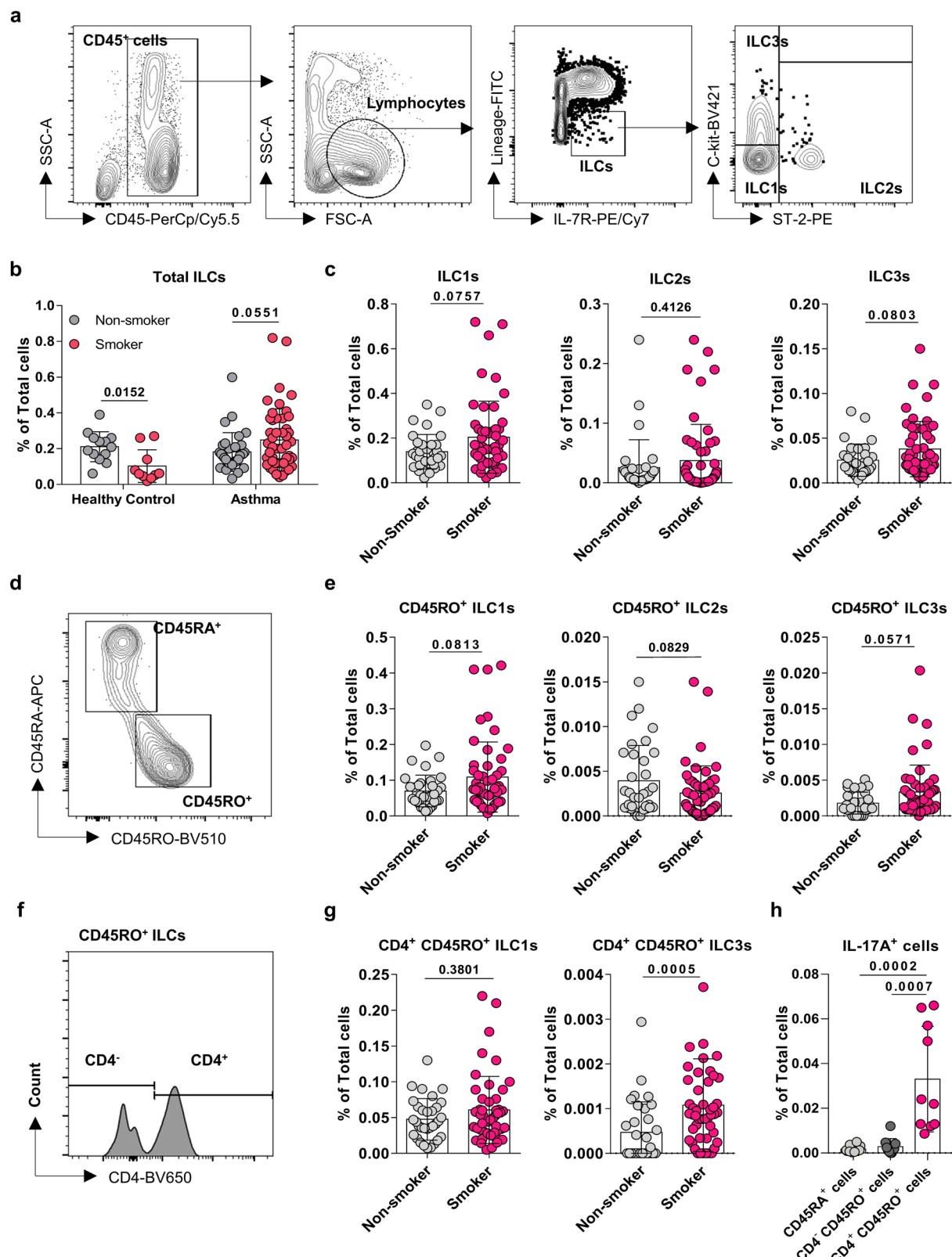

(Fig. 3h). Thus, in asthma patients but not healthy controls, CSE treatment readily activated ILC3s in the blood into memory-like CD45RO$^+$ILC3s. These data together suggest that cigarette smoke directly induces memory-like CD45RO expressing ILC3s (but not ILC1s) only in patients with asthma.

**Cigarette smoke-induced damage to airway epithelial cells generate an ILC3-activating environment.** Cigarette smoke can damage airway epithelial cells, thereby increasing mucosal permeability and the infiltration of inflammatory immune cells[6,21]. To examine whether CSE affects the barrier function of airway

**Fig. 2 Smoking in asthma associates with increased peripheral blood CD4$^+$CD45RO$^+$ILC3 frequencies. a** Gating strategy for the three subsets of innate lymphoid cells (ILCs; Lineage$^-$IL-7R$^+$) in the blood. **b** Comparison of non-smoking and smoking asthma patients and non-smoking and smoking healthy individuals in terms of total ILC frequencies in the peripheral blood. Comparison of non-smoking and smoking asthma patients in terms of peripheral blood ILC1 (ST-2$^-$C-kit$^-$), ILC2 (ST-2$^+$), and ILC3 (ST-2$^-$C-kit$^+$) (**c**), CD45RO$^+$ ILC1, ILC2, and ILC3 (**d**, **e**), and CD4$^+$CD45RO$^+$ILC1 and CD4$^+$CD45RO$^+$ILC3 (**f**, **g**) frequencies. **h** Frequencies of CD45RA$^+$, CD4$^-$CD45RO$^+$, and CD4$^+$CD45RO$^+$ total ILCs that co-express IL-17A. Each dot represents individual subject. Sample size of non-smoking healthy controls, $n = 13$; smoking healthy controls, $n = 10$; non-smoking asthma patients, $n = 32$; smoking asthma patients, $n = 49$ for Fig. 2b, non-smokers, $n = 33$; smokers, $n = 52$ for ILC1s, ILC2s in Fig. 2c and CD45RO$^+$ILC1s, CD45RO$^+$ILC2s in Fig. 2e, non-smokers, $n = 32$; smokers, $n = 52$ for ILC3s in Fig. 2c and CD45RO$^+$ILC3s in Fig. 2e, non-smokers, $n = 32$; smokers, $n = 49$ for CD4$^+$CD45RO$^+$ILC1s and non-smokers, $n = 32$; smokers, $n = 48$ for CD4$^+$CD45RO$^+$ILC3s in Fig. 2g, CD45RA$^+$ cells; $n = 10$, CD4$^-$CD45RO$^+$ cells; $n = 10$, CD4$^+$CD45RO$^+$ cells; $n = 10$ for Fig. 2h. The non-smokers and smokers in the asthma patients or healthy individuals were compared by multiple t-tests (Fig. 2b) or two-tailed Mann-Whitney $U$ test (Fig. 2c, e, g). One-way ANOVA was conducted to compare the frequencies of IL-17A$^+$ cells among CD45RA$^+$, CD4$^-$CD45RO$^+$, and CD4$^+$CD45RO$^+$ ILCs (Fig. 2h). All data are presented as mean ± standard deviation. $p < 0.05$ is considered as significant.

epithelial cells, we treated the A549 human alveolar basal epithelial cell line with CSE. The treatment decreased A549 expression of E-cadherin (which associates with epithelial tight junction functions[22]) and destroyed the close-knit epithelial layer (Fig. 4a). To further assess the effect of CSE on epithelial barrier function, we treated the A549 cell line, the RPMI2650 human nasal epithelial cell line, the BEAS-2B human bronchial epithelial cell line, and the MLE12 mouse lung epithelial cell line with CSE and measured their transepithelial electrical resistance (TEER). CSE significantly decreased TEER in all lines (Fig. 4b).

Since damaged epithelial cells can release alarmins such as IL-1, IL-23, and IL-33, all of which are essential for ILC activation[10,14], we examined the effect of CSE treatment on A549 expression of these innate cytokines. CSE treatment elevated *IL1B* expression in a dose-dependent manner, decreased *IL23A* expression at the highest dose, and had no effect on *IL33* expression (Fig. 4c). Increasing the duration of CSE treatment also augmented the IL-1β mRNA and protein levels in the A549 cells (Supplementary Fig. 7a). In addition, CSE treatment elevated *IL1B* expression in the other epithelial cell lines (Fig. 4d). However, the expression of *IL23A* and *IL33* did not differ according to CSE treatment with the exception of RPMI2650 (Supplementary Fig. 7b, c).

Since IL-1β engages an IL-1 receptor on ILC3s and promotes their proliferation and IL-17A production[14], these findings suggest that the smoking-induced sputum ILC3s and circulating memory-like CD4$^+$CD45RO$^+$ILC3 subset in asthma patients could be generated by smoke-damaged airway epithelial cells rather than circulating immune cells. Notably, the whole PBMCs or circulating immune cell subsets from the four healthy and asthmatic smoker/never-smoker groups did not differ in terms of IL-1β-producing cell frequencies (Supplementary Fig. 8a–c): this analysis supports the notion that the IL-1β potentially driving the smoking-induced generation of ILC3s in asthma is produced locally rather than systemically.

**The increased ILC3 frequencies in asthma patients correlate with asthma severity.** To understand the clinical significance of the increased ILC3 frequencies in smoking asthma patients, we analyzed the relationships between the frequencies of sputum ILC3s and blood CD4$^+$CD45RO$^+$ILC3s and asthma severity, as measured by (i) the pulmonary function indices FEV$_1$ and FEV$_1$/FVC ratio or (ii) asthma control, as measured by the ACQ and ACT scores. ACQ and ACT quantify asthma control and play a prominent role in Global Initiative for Asthma guidelines[23]:patients with well-controlled asthma have higher ACT and lower ACQ values. ILC3 frequencies in the sputum of asthma patients correlated negatively with FEV$_1$ (Fig. 5a), FEV$_1$/FVC ratio (Fig. 5b), and ACT score (Fig. 5c) and positively with ACQ score (Fig. 5d). The CD4$^+$CD45RO$^+$ILC3 frequencies in the peripheral blood also correlated negatively with the pulmonary function indices but not the asthma control indices (Fig. 5e–h). Thus, the frequencies of

ILC3 subsets in the induced sputum and blood of asthma patients correlated with their disease severity.

**The increased ILC3 frequencies in smoking asthma patients associate with a non-Th2 asthma phenotype.** Asthma is a heterogeneous disease that can be divided into eosinophilic (type 2 immunity-dominant) and non-eosinophilic (type 1/3 immunity-dominant) asthma. Some forms of non-eosinophilic asthma also associate with neutrophilia[24]. Therefore, we asked whether the increased ILC3 frequencies in asthma patients correlated with their circulating eosinophil and neutrophil counts. The sputum ILC3s and circulating CD4$^+$CD45RO$^+$ILC3s correlated positively with circulating neutrophils (Fig. 6a, b). Despite the fact that the smoker and non-smoker asthma patients had similar numbers of circulating neutrophils, circulating neutrophils correlated positively with smoking amount (PY) (Supplementary Fig. 9a, b). By contrast, circulating eosinophil counts did not correlate with either blood/sputum ILC3 frequencies and negatively correlated with smoking amount (PY) (Fig. 6c, d, and Supplementary Fig. 9c, d). Since blood/sputum ILC3 frequencies also correlated with smoking amount (Fig. 3a, b), impaired lung function (Fig. 5), and neutrophil counts (Fig. 6a, b), these findings suggest that smoking-upregulated ILC3 subsets in asthma patients associate with neutrophilic but not eosinophilic immune responses that are harmful to lung function.

We previously reported that ILCs regulate macrophage polarization in patients with asthma: ILC1 and ILC3 associate with classical macrophage (M1) polarization and non-eosinophilic asthma, whereas ILC2 associates with alternative macrophage (M2) polarization and eosinophilic asthma[13]. Therefore, we also analyzed the macrophages in the induced sputum of the smoking and non-smoking asthma patients by flow cytometry and then determined their relationship with ILC3 frequencies. Thus, total macrophages were gated as CD45$^+$CD68$^+$ cells and then divided into M1 (CD11c$^+$CD206$^-$) and M2 (CD11c$^-$CD206$^+$) macrophages (Supplementary Fig. 9e). Smoking in asthma associated with elevated total and M1 type macrophage frequencies but a similar association with M2 type macrophage frequencies was not observed (Supplementary Fig. 9f, g). Moreover, sputum ILC3 frequencies correlated positively with M1 (Fig. 6e), but not M2 (Fig. 6f) macrophage frequencies. Blood CD4$^+$CD45RO$^+$ILC3 frequencies did not correlate with either M1 or M2 macrophage frequencies (Fig. 6g, h). Notably, sputum M1 macrophage frequencies did not associate with impaired lung function, unlike neutrophil counts (Supplementary Fig. 10a–d). This suggests that neutrophils interact more closely with ILC3 subsets in the pathogenesis of smoking asthma than M1 macrophages.

Taken together, these data suggest that the cigarette smoking-related increase in ILC3 frequencies that are seen in asthma patients may reflect a non-eosinophilic phenotype of asthma that is characterized by non-Th2 type inflammation such as M1 macrophages and neutrophils.

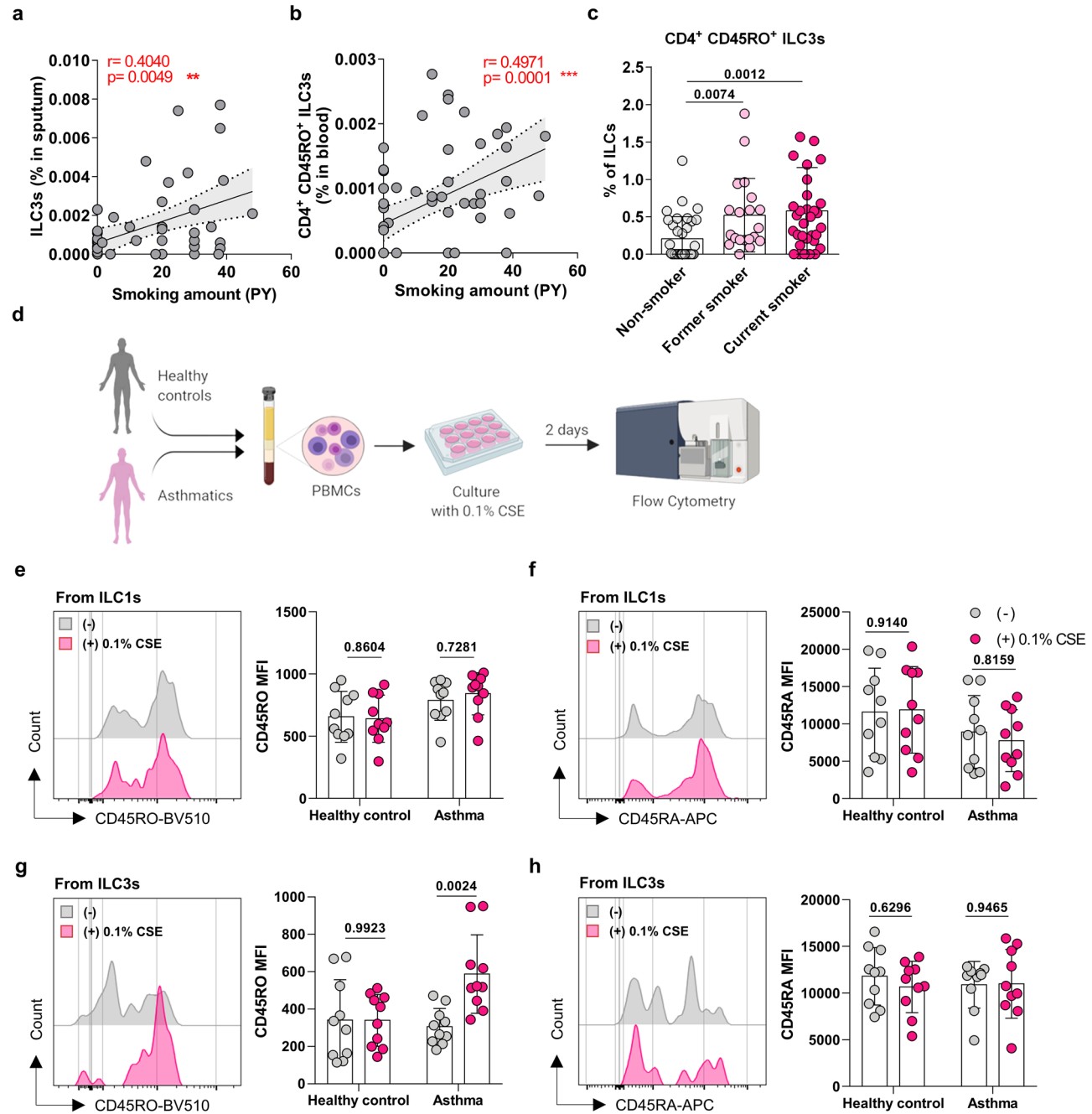

**Fig. 3 In vitro cigarette smoke exposure elevates the CD45RO expression of ILC3s from asthma patients but not healthy individuals.** Correlation between smoking amount (PY) and (**a**) ILC3 frequency in induced sputum and (**b**) CD4+CD45RO+ILC3 frequencies in peripheral blood, as determined by Spearman $r$ correlation test. **c** Comparison of non-smokers, former smokers, and current smokers in asthma patients in terms of CD4+CD45RO+ILC3s in the peripheral blood. PBMCs were treated in vitro with cigarette smoke extract (CSE) as shown schematically in (**d**). Healthy individuals and asthma patients were compared in terms of the CD45RO (**e, g**) and CD45RA (**f, h**) mean fluorescence intensity (MFI) of ILC1s (**e, f**) and ILC3s (**g, h**) that were and were not treated with 0.1% CSE. Each dot represents individual subject. Sample size of non-smokers, $n = 33$; former smokers, $n = 20$; current smokers, $n = 32$ for Fig. 3c, each healthy control with or without 0.1% CSE treated, $n = 10$; each asthma patient with or without 0.1% CSE treated, $n = 10$ for Fig. 3e–h. Correlation analyses were conducted with Spearman $r$ correlation test, dotted line represents 95% confidence interval (Fig. 3a, b). Comparison of the CD4+CD45RO+ILC3s frequencies among non-smokers, former smokers, and current smokers in asthma patients was conducted by one-way ANOVA (Fig. 3c). CSE-treated and untreated cultures were compared by multiple $t$ tests (Fig. 3e–h). All data are presented as mean ± standard deviation. $p < 0.05$ is considered as significant. Figure 3d was created with Biorender.com.

## Discussion

Many clinical, functional, and pathological studies have been conducted on the effect of cigarette smoking on COPD and other smoking-induced airway diseases. By contrast, little is known about the relationship between active smoking and asthma. The scanty data that do exist suggest that exposure to cigarette smoke aggravates asthma symptoms[25] and increases asthma-related mortality[26]. Moreover, the intensity of smoking associates with the risk of new-onset asthma in allergic adults[26]. However, the mechanisms by which cigarette smoke worsens asthma are unclear. This hampers

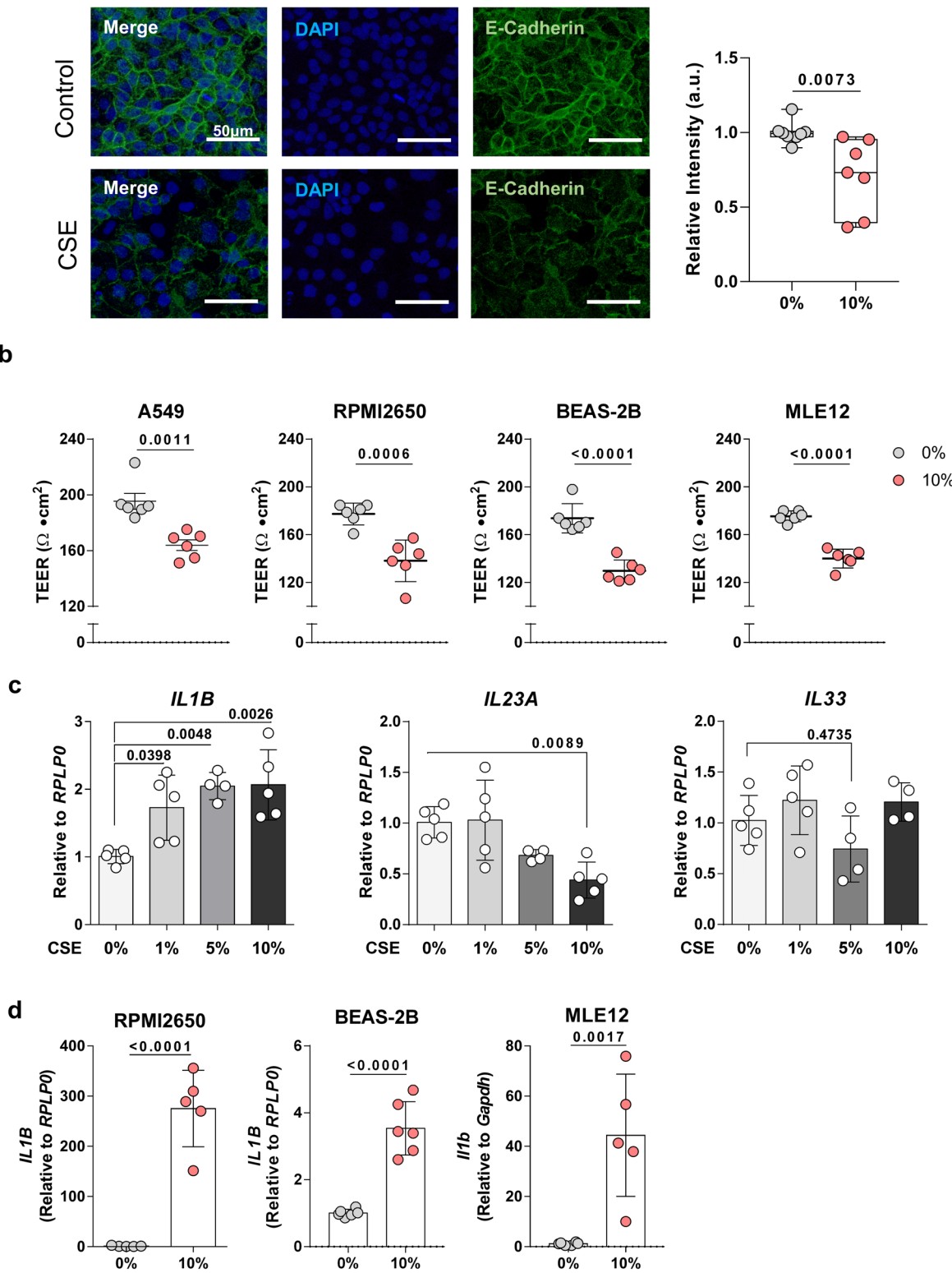

the development of treatment options for smoking-related asthma disease. We demonstrated here for the first time that compared to non-smoking asthma patients, smokers with asthma had increased frequencies of ILC3s in the airway along with increased frequencies of memory-like ILC3s (CD4+CD45RO+ILC3s) in the peripheral blood. Significantly, these alterations correlated with increased smoking amount and decreased lung function. Thus, smoking may worsen asthma by elevating ILC3 frequencies in the airway and peripheral blood. These findings are supported by several studies that suggest ILC3s are linked to smoking. First, COPD patients have increased NKp44−ILC3(NCR−ILC3) frequencies in their blood compared to control groups[27,28]. Second, the lungs of cigarette smoke-exposed mice have elevated IL-17A- or IL-22-expressing ILC3 frequencies[29].

**Fig. 4 Cigarette smoke exposure promotes airway epithelial cell damage and IL-1β secretion. a** The human alveolar epithelial cell line A549 was treated with 10% cigarette smoke extract (CSE) for 48 h and then subjected to anti-E-cadherin immunostaining (AF488; green) and DAPI staining (blue). Scale bar, 50 μm. The plot shows the relative intensity of E-Cadherin. **b** A549 cells, the human airway epithelial cell lines RPMI2650 and BEAS-2B, and the mouse airway epithelial cell line MLE12 were treated with 10% CSE for 48 h and their transepithelial electrical resistance (TEER) was measured. **c** A549 cells were treated with 0–10% CSE for 48 h and their *IL1B*, *IL23A*, and *IL33* mRNA expressions were measured. **d** Comparison of *IL1B* expressions from CSE treated and untreated RPMI2650, BEAS-2B, and MLE12. Sample size of 0% CSE treated A549, *n* = 8; 10% CSE treated A549, *n* = 7 for Fig. 4a, 0% CSE treated, *n* = 6; 10% CSE treated, *n* = 6 for A549, RPMI2650, BEAS-2B, and MLE12 in Fig. 4b, 0% CSE treated, *n* = 5; 1% CSE treated, *n* = 5; 5% CSE treated, *n* = 4; 10% CSE treated, *n* = 5 for *IL1B*, *IL23A* analysis in Fig. 4c, 0% CSE treated, *n* = 5; 1% CSE treated, *n* = 5; 5% CSE treated, *n* = 4; 10% CSE treated, *n* = 4 for *IL33* analysis in Fig. 4c, 0% CSE treated, *n* = 5; 10% CSE treated, *n* = 5 for RPMI2650 and BEAS-2B in Fig. 4d, 0% CSE treated, *n* = 5; 10% CSE treated, *n* = 4 for MLE12 in Fig. 4d. Each of the data was the representative data from more than twice replications. For the box plots, lower and upper box boundaries 25th and 75th percentiles, respectively, line inside box median, lower and upper error lines 10th and 90th percentiles, dots indicate data points, respectively. Relative intensity of E-cadherin, TEERs, and gene expressions from CSE-treated and untreated cell lines were compared by two-tailed unpaired *t* test (Fig. 4a, b, d). The CSE-treated cells were compared to the untreated cells in different CSE concentrations by one-way ANOVA (Fig. 4c). All data are presented as mean ± standard deviation. *p* < 0.05 is considered as significant.

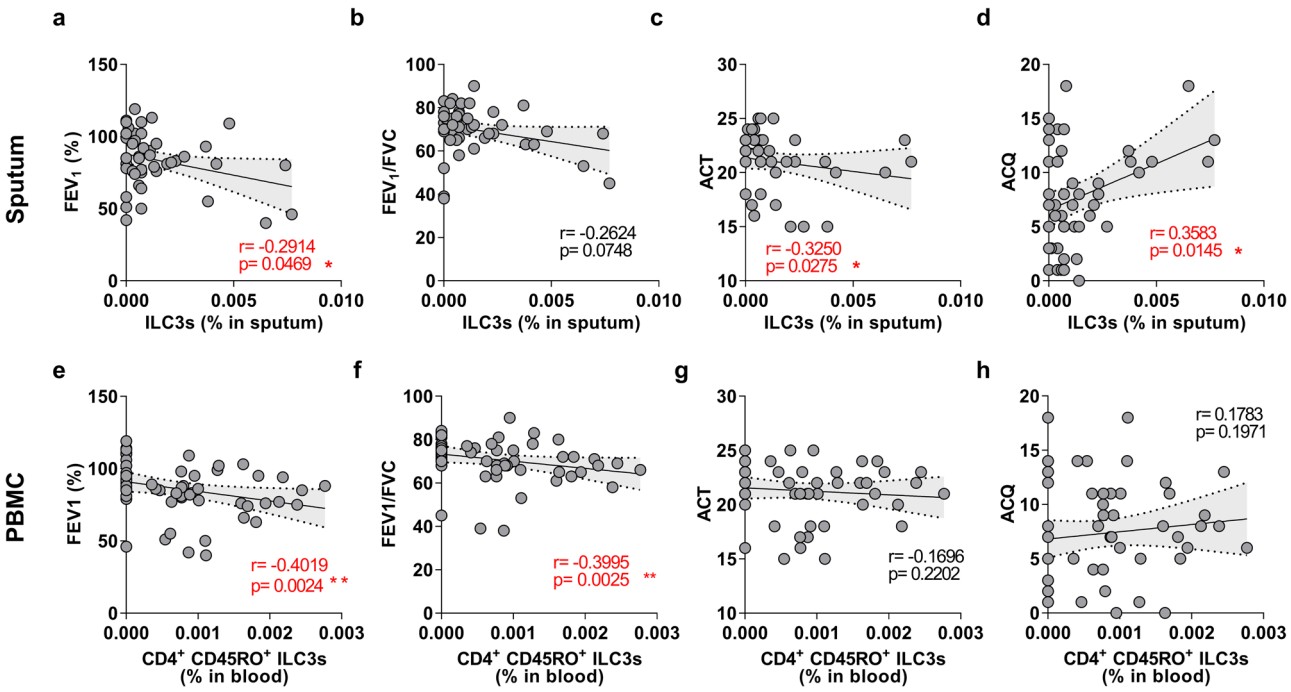

**Fig. 5 ILC3s in sputum and CD4+CD45RO+ILC3s in blood correlate positively with asthma severity.** Correlation between ILC3 frequencies in induced sputum and four clinical indices of asthma severity, namely, FEV₁ (%) (**a**), FEV₁/FVC (**b**), ACT (**c**), and ACQ (**d**). Correlation between CD4+CD45RO+ILC3 frequencies in the blood and FEV₁ (%) (**e**), FEV₁/FVC (**f**), ACT (**g**), and ACQ (**h**). Each dot represents individual subject. Correlation analyses were conducted with Spearman *r* correlation test, dotted line represents 95% confidence interval. *P < 0.05, **P < 0.01.

The notion that the adverse consequences of smoking in asthma may be due to altered immune responses in the airway was initially advanced by studies showing that cigarette smoke causes critical changes in the macrophages and neutrophils of asthma patients[30]. In particular, cigarette smoke was found to induce alveolar macrophages and neutrophils to produce pro-inflammatory cytokines, reactive oxygen species, matrix metalloproteinases, and various chemokines[31]. Indeed, we observed that smoking in asthma associated with elevated frequencies of pro-inflammatory M1 macrophages in the sputum. However, M1 macrophages did not associate with decreased lung function. By contrast, we did observe that circulating neutrophils correlated with both smoking amount and decreased pulmonary function. Given that the elevated ILC3 frequencies in smokers with asthma correlated positively with not only smoking amount and decreased lung function but also M1 macrophage and circulating neutrophil frequencies, it may be the key player in the pathogenesis of smoking-related asthma is the ILC3s.

It should be noted that another ILC subtype, ILC2s, is thought to play an important role in allergic asthma: these cells help to orchestrate and propagate type 2 inflammation in the airways[32]. However, two studies have suggested that tobacco smoke directly inhibits the function of ILC2s, thereby affecting the pathogenesis of two respiratory diseases, namely, asthma and COPD. One study showed that murine and human ILC2s express the α7 nicotinic acetylcholine receptor, a receptor for the tobacco component nicotine, and that nicotine agonists inhibit ILC2 cytokine production by decreasing Gata3 expression and dampening NF-kB pathway activation[33]. The other study reported that chronic exposure to cigarette smoke silences ILC2s by regulating the IL-33-ST2 axis[34]. These findings appeared to contradict the notion that smoking is a risk factor for asthma.

In the current study, we found that cigarette smoke exposure in patients with asthma changed ILC3s rather than ILC2s. Our experiments also suggested that cigarette smoke may affect ILC3s both directly and indirectly. The indirect effects were shown by

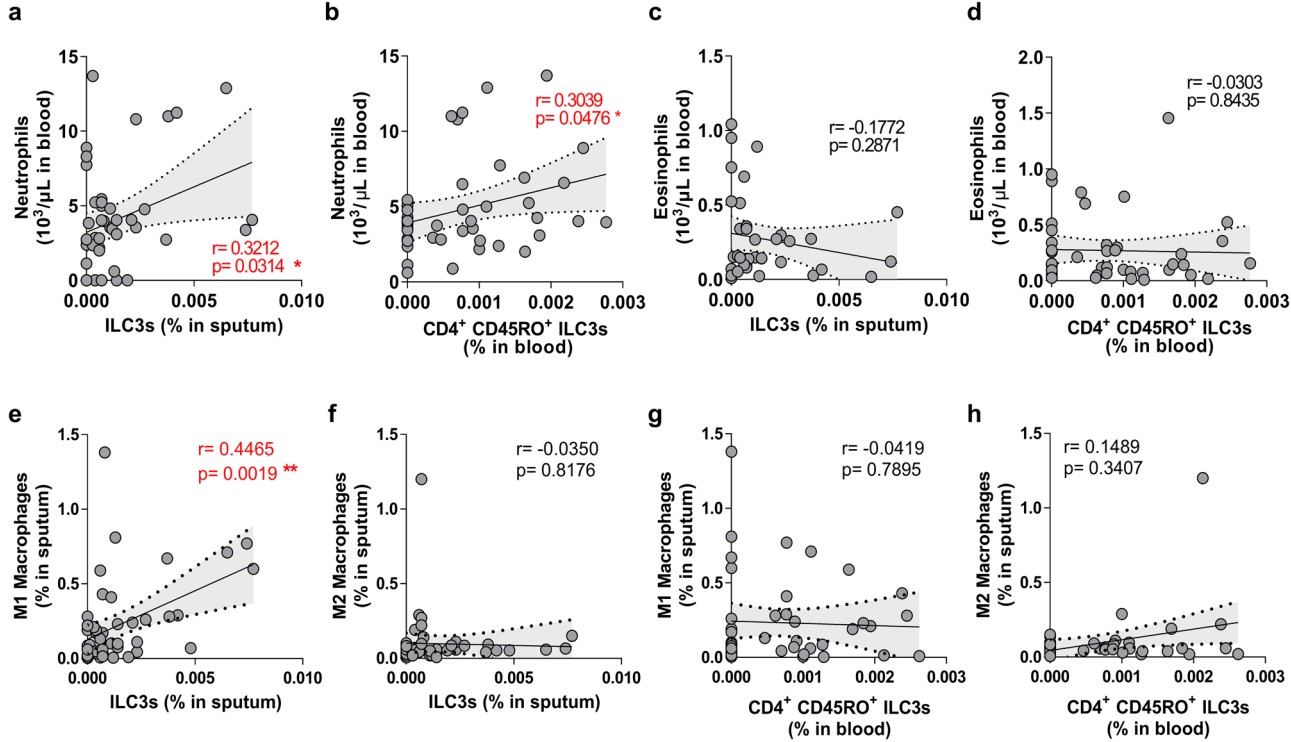

**Fig. 6 Sputum ILC3 and circulating CD4⁺CD45RO⁺ILC3 frequencies in asthma patients correlate with circulating neutrophil counts and sputum M1 macrophage frequencies.** Correlations between ILC3 frequencies in induced sputum (**a**, **c**, **e**, **g**) or CD4⁺CD45RO⁺ILC3 frequencies in the blood (**b**, **d**, **f**, **h**) and circulating neutrophil counts (**a**, **b**), circulating eosinophil counts (**c**, **d**), sputum M1 macrophage frequencies (**e**, **f**), or sputum M2 macrophage frequencies (**g**, **h**). Correlation analyses were conducted with Spearman $r$ correlation test, dotted line represents 95% confidence interval. *$P < 0.05$, **$P < 0.01$.

treating airway epithelial cells with CSE: this exposure caused these cells to release IL-1β in a CSE-dose and culture time-dependent manner. IL-1β is an important cytokine for ILC3s since it promotes their differentiation and activation[14]. Notably, Wang et al. reported that cigarette smoke exposure also induces airway epithelial cells to produce CCL20, a chemoattractant for CCR6-expressing cells[35], which include ILC3s[36]. Thus, the findings of Wang et al. and our study suggest that smoking may generate an environment that promotes the airway retention of ILC3s and their activation, respectively. The direct effects of cigarette smoke on ILC3s that we observed were shown by treating PBMCs with CSE: this treatment upregulated ILC3 expression of CD45RO in the asthma patients but not the healthy individuals. It remains at present unclear how ILC3s recognize CSE. Cigarette smoke contains more than 5000 chemicals that are toxic to our body, including our immune system[37]: for example, they increase reactive oxygen species production and modulate immune functions[7], while nicotine, which is abundant in CSE, suppresses T cells by inducing anergy and apoptosis[38,39]. It is possible that the tar, nicotine, and carbon monoxide in cigarette smoke[40] activate ILC3s by binding to toll-like receptors and aryl hydrocarbon receptors[41].

Our study showed not only that CSE generates CD45RO⁺ILC3s in vitro but also that smokers with asthma had greater frequencies of CD45RO⁺ILC3s in their peripheral blood. CD45 is one of the most abundant glycoproteins on the surface of immune cells; it also has phosphatase activity that suppresses signal transduction such as that by JAK and LCK[42,43]. Since CD45RO has less phosphatase activity than CD45RA[44], the transition of CD45RA to CD45RO suggests that immune cells have become more sensitive to external stimuli[45]. This isoform switch is well observed during T cell activation: antigen-naïve T cells express CD45RA whereas activated/memory T cells express CD45RO[19]. Recently, it was reported that

ILCs can also express CD45RO on their surface, which suggests they have been primed[17,18]. In line with our observation that smoking asthma patients had higher frequencies of blood CD45RO⁺ILC3s, Shikhagie et al. showed that compared to control subjects, smokers and patients with COPD have higher frequencies of lung NRP1⁺ILC3s that highly express CD45RO; moreover, those cells were observed to secrete more cytokines than NRP1⁻ILC3s[17]. Thus, cigarette smoke may increase the peripheral blood frequencies of CD45RO⁺ILC3 cells that are highly primed to produce inflammatory cytokines.

In conclusion, this study showed for the first time that the ILC3s in both the airways and peripheral blood of smoking asthma patients undergo marked smoking-specific changes that correlate significantly with asthma severity. These data also suggest that cigarette smoke may mediate some of its adverse effects on asthma by altering airway immunity toward non-Th2 type inflammation. Our findings are supported by the fact that ILC3 gene signatures are highly enriched in adult-onset asthma, which is often more severe than childhood-onset asthma and associates with a worse prognosis[46]. Severe asthma and steroid-resistant asthma also associate with elevated IL-17A levels in the lung tissue[47]. The potential importance of ILC3s in asthma is also supported by the fact that ILC3s are the most abundant ILCs in the lungs of healthy individuals under homeostatic conditions[28]. It should be noted that our study largely focused on associations between smoking and immune cells in patients with asthma; consequently, it cannot yet be concluded that ILC3 cells play a prime role in asthma pathogenesis. However, if our findings are supported by additional clinical and experimental studies, they may open the door to efficient and personalized treatment regimens that target pulmonary ILC3s and could help manage asthma exacerbations caused by smoking. Our present study also suggests that sputum ILC3s and blood CD45RO-

expressing ILC3s could serve as possible biomarkers for asthma severity in smokers.

## Methods

**Participants**. We recruited 33 non-smokers with asthma (non-smoker; who have never smoked), 58 smokers with asthma who are currently smoking or have smoked previously, 13 non-smoking healthy controls, and 11 healthy smoking controls from Seoul National University Hospital (Seoul, South Korea) and Chung-Ang University Hospital (Seoul, South Korea) between December 2016 and June 2017. All patients with asthma met one of the following criteria: $FEV_1$ changed over 12% and 200 mL after bronchodilator response, and/or there was significant airway hyperresponsiveness to methacholine or mannitol provocation (Table 1). Patients with cancer, severe medical conditions, or other pulmonary diseases were excluded along with patients on medications such as antibiotics, antifungal agents, antiviral drugs, probiotics, or any systemic steroids. Patients treated with immunotherapy were also excluded. We checked and followed the Strengthening the Reporting of Observational studies in Epidemiology (STROBE) guidelines.

**Study approval**. All participants enrolled in this study provided written informed consent. The study protocol was approved by the Chung-Ang University Hospital Institutional Review Board (IRB number 1600-002-253) and the Seoul National University Hospital Institutional Review Board (IRB number 1608-163-788).

**Immune cell isolation from induced sputum and peripheral blood**. To eliminate mucus from sputum samples, the same volume of 0.1% dithiothreitol (Sigma, MO, USA) was added, the tube was shaken for 20 min at 37 °C, and the mixture was filtered through a 70 µm strainer. After centrifugation at 1400 rpm for 6 min, the cell pellet was resuspended with 100 µL FACS buffer (PBS containing 2% fetal bovine serum) for staining. Peripheral blood (PB) mononuclear cells (PBMC) were isolated by using Ficoll-Paque PLUS density gradient media (GE Healthcare, IL, USA). Briefly, PB was centrifuged at 2000 rpm for 10 min at 4 °C to separate the plasma from the cells. After removing the plasma, the remaining cell pellet was resuspended with PBS and loaded onto a Ficoll-Paque layer. After centrifuging at 1800 rpm for 30 min, the PBMC layer was collected and washed with PBS.

**PBMC stimulation**. To stimulate PBMCs, they were cultured in RPMI 1640 containing 10% FBS and 10 µg/mL gentamycin. 1 µg/mL ionomycin and 100 ng/mL phorbol 12-myristate 13-acetate were added to the culture media with 0.01% Protein Transport Inhibitor (BD GolgiStopTM, BD biosciences, NJ, USA). After 3 h at 37 °C, the PBMCs were harvested.

**Flow cytometry analysis**. Cells isolated from induced sputum and PBMCs were blocked with anti-CD16/CD32 and stained with the following fluorochrome-labeled monoclonal antibodies: anti-CD45 (HI30; BD Biosciences, NJ, USA); anti-CD3ε (UCHT1), anti-CD11c (3.9), anti-CD11b (ICRF44), anti-CD14 (HCD14), anti-CD19 (HIB19), anti-CD49b (P1E6-C5), and anti-FcεRIα (AER-37), which were used as lineage markers and were from BioLegend (CA, USA); anti-CD68 (Y1/82A), anti-CD117 (C-Kit, 104D2), anti-CD127 (IL-7R, A019D5), anti-CD206 (15-2), anti-HLA-DR (L243), anti-CD4 (OKT4), anti-CD45RO (UCHL1), anti-CD45RA (HI100), anti-CD56 (HCD56), anti-CD16 (3G8), anti-NKp44 (P44-8), anti-IFNγ (4S.B3), anti-IL-5 (TRFK5), anti-IL-17A (BL168), and anti-IL-1β (H1b-98), which were from BioLegend (CA, USA); and anti-ST-2 (B4E6; MD Bioproducts, MN, USA). To analyze cytokine production, PBMCs were fixed and permeabilized with the Fixation/Permeabilization Solution Kit (BD CytoFix/CytoPermTM, BD biosciences, NJ, USA). Flow cytometry was performed using BD LSRFortessa™ and BD LSRFortessa™ X-20 (BD, NJ, USA) and analyzed by FlowJo (V10) software (BD, NJ, USA). Detailed information about the antibodies is provided in Supplementary Table 1.

**Preparation of cigarette smoke extract**. Cigarette smoke extract (CSE) was generated from a single pack of Marlboro Red cigarettes (Philip Morris, VA, USA) containing 8 mg tar and 0.7 mg nicotine. The smoke from this pack of cigarettes was dissolved in 20 mL PBS (considered to be 100% CSE) by using a vacuum pump. After dissolving the cigarette smoke, CSE was sterilized by filtration through a 0.2 µm filter.

**Cigarette smoke extract treatment**. To analyze the direct effects of CSE on ILCs, PBMCs isolated from healthy controls and asthma patients were cultured for 48 h with RPMI1640 containing 10% FBS and 0.1% of CSE. The cells were then harvested and analyzed by flow cytometry.

The effect of CSE treatment on epithelial cells was assessed with the human alveolar basal epithelial cell line A549, the human nasal epithelial cell line RPMI2650, the human bronchial epithelial cell line BEAS-2B, or the mouse lung epithelial cell line MLE12. Thus, $1 \times 10^5$–$5 \times 10^5$ cells were seeded for 48 h and then cultured with 1, 5, or 10% CSE for 6–48 h. To analyze IL-1β secretion, 5 mM ATP (Sigma, MO, USA) was added to the culture media for the last 6 h before harvest.

**Confocal microscopy to measure airway epithelial cell layer integrity**. To determine the effect of CSE on epithelial intercellular adhesion, coverslips were coated with poly-L-lysine solution (Sigma, MO, USA) for 5 min at RT, washed with PBS, and dried for 1 h at 60 °C. Coated coverslips were loaded into a culture plate and cultured with A549 cells as described above for 48 h. The coverslips were then washed with PBS and the cells were fixed with 4% PFA for 15 min, washed with 0.05% Tween 20 (Promega, WI, USA) diluted in PBS (PBST), permeabilized with 0.2% Triton X-100 (Promega, WI, USA) for 10 min, and blocked with 3% BSA in PBST for 3 h. The cells were stained for 2 h at RT with anti-human E-cadherin antibody (Invitrogen, CA, USA) that was diluted in PBST containing 1% BSA. The cells were then incubated with Alexa Fluor 488-labeled anti-mouse IgG antibody (Thermofisher, MA, USA) for 1 h. The coverslips were flipped over onto ProLong diamond antifade mountant with DAPI (Invitrogen, CA, USA)-loaded slide glasses and incubated overnight at 4 °C. The slides were imaged with a Confocal-A1 (Nikon, Tokyo, Japan) confocal microscope and analyzed by Image J software (NIH, MD, USA). Detailed information about the antibodies is provided in Supplementary Table 1.

**TEER measurements of airway epithelial cell layer integrity**. TEER measurements were conducted with an Epithelial Voltohmmeter (World Precision Instruments, FL, USA). A549, RPMI2650, BEAS-2B, or MLE12 cells were seeded on 12 well-transwell inserts (Corning, ME, USA) for 24 h before CSE treatment and then, cells were cultured with 10% CSE-containing media for 48 h, rinsed with PBS. The electrodes were swiped with 70% ethanol and rinsed with PBS prior to use. TEER was calculated by using the following equation[48]: TEER $(\Omega \, cm^2) = [R_{total} (\Omega) - R_{blank} (\Omega)] \times$ membrane area $(cm^2)$.

**RT-qPCR**. The total RNA of cultured epithelial cell lines was extracted by using TRIzol (Invitrogen, CA, USA) reagent according to the manufacturer's protocol. cDNA was synthesized with the iScript cDNA Synthesis Kit (Bio-Rad, CA, USA). RT-qPCR assay was performed with IQ Supermix (Bio-Rad, CA, USA). Primers of IL1B, IL23A, and IL33 were purchased from Thermofisher Scientifics (TaqMan Gene Expression Assay; Thermofisher, CA, USA).

**ELISA**. ELISA of supernatant IL-1β (BD Biosciences, NJ, USA) was conducted according to the instructions of the manufacturer.

**Statistics**. The data are presented as mean ± standard deviation. To determine the normality of data, Shapiro-Wilk normality tests were conducted. Two groups were compared by using two-tailed Mann-Whitney $U$ test or two-tailed unpaired $t$-test. Three or more groups were compared with one-way ANOVA following by a Bonferroni's post-test or two-way ANOVA. All correlation analyses were conducted with Spearman $r$ correlation test. $P$ values of <0.05 were considered to indicate statistical significance. GraphPad Prism 7 was used for statistical analysis.

**Reporting summary**. Further information on research design is available in the Nature Research Reporting Summary linked to this article.

## Data availability

Source data are provided in this paper. The source data underlying all reported averages in graphs underlying Figs. 1b–f, 2b, c, e, g, h, 3a–c, e–h, 4a–f, 5a–h, 6a–h and Supplementary Figs. 1–10 are provided as a Source data file.

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

## Acknowledgements

J.H. received a scholarship from the BK21 FOUR education program. This research was supported by a grant (code: 2016-ER6705-00) from the Research of Korea Centers for Disease Control and Prevention to J.W.J.; by grants from the Korea Healthcare Technology R&D Project of the Ministry of Health and Welfare, Korea (HI15C3083) to H.Y.K.; and a grant from the National Research Foundation of Korea (SRC 2017R1A5A1014560, NRF-2019R1A2C2087574 and NRF-2022R1A2C3007730) to H.Y.K. Cartoon in Fig. 3d was created with BioRender.com.

## Author contributions

J.H. and J.K. carried out the experiments and analyzed the data. J.H., J.K., and H.Y.K. wrote the manuscript. K.H.S., I.W.P., and B.W.C. were involved in recruiting the subjects and collecting the samples and clinical data. D.H.C., S.H.C., and H.R.K. contributed to the interpretation of the results. J.W.J. and H.Y.K. supervised the project.

## Competing interests

The authors declare no competing interests.
