## [Peer Review File · Nature Communications]

Cigarette smoke aggravates asthma by inducing memory-like type 3 innate lymphoid cellsEditorial Note: Parts of this Peer Review File have been redacted as indicated to remove third-party material where no permission to publish could be obtained.

REVIEWER COMMENTS

Reviewer #1 (Remarks to the Author):

This study examines the potential role of ILC3s in smoking induced asthma. The link between ILC2s and allergic asthma has been extensively studied but the potential role of ILC3s is less well established. This study takes several novel approaches, including the quantification of ILC3s in sputum samples, a convenient method of sampling airway cells. Differences in the frequency of certain ILC3 phenotypes between smoking and non-smokers with and without asthma and the (weak) correlations with measures of smoking frequency and disease severity do suggest potentially meaningful differences between these groups have been identified. These findings are novel and of general interest. The paper is very clearly written and the figures are well presented and clear.

However, my underlying concern with this study is whether or not the cells identified are in fact ILCs and not some other cell type. Given that ILCs are generally extremely low frequency, even low level contamination from, say, T-cells expressing low levels of CD3 can have a huge impact. Several other factors add to the uncertainty around whether or not bona fide ILCs are being measured:

(1) The frequency of ILCs identified in PBMCs are highly inconsistent with other reports from humans, in which ILC2s are reported as the dominant ILC class, followed by ILC3 precursors and low levels of ILC1s. By contrast supplementary fig.3 suggest ILC1s are by far the dominant ILC class (median frequency approximately 0.175% of total cells, in contrast to less than 0.001% generally reported), followed by ILC3s (here approximately 0.02%, which is consistent) and then extremely low levels of ILC2s (here approximately 0.005%, which is at least an order of magnitude less than expected). As ILC2s are always reported as the most frequent ILC cell type in the blood of healthy controls this does not give any confidence in the identification of ILCs here.

(2) The identification of CD45RO+ve ILC3s in the blood. The authors justify this with reference to a paper by Shikhagaieer et al. showing elevated CD45RO expressing ILC3s in blood of individuals with COPD. However, I cannot find any evidence of this data in the reference given. The authors do observe expression of CD45RO on NRP1+ ILC3s in lung tissue and in tonsils, but I find no data on CD45RO expression on blood ILC3s. The study does, however, present clear evidence that blood ILC3 do not express NRP1, and in tissue CD45RO is only expressed on NRP1+ve ILC3s. Therefore I do not understand the rationale for using this marker, especially when other studies (doi: 10.4049/jimmunol.1501491) suggests that CD45RO is expressed on ILC1s in human PBMC but not on ILC2s and ILC3s.

(3) Expression of CD4 on circulating human ILC3s. In addition to the apparently unusual expression of CD45RO on circulating ILC3s, the biggest differences are observed when these cells are divided into CD4+ and CD4- ILC3s. However, I am not aware of other publications that have identified CD4 expression on circulating human ILC3s. The following publication, for example, identifies potential expression on ILC1-like cells but finds no expression on ILC3s (doi: 10.4049/jimmunol.1501491).

Both points (2) and (3) may potentially reflect disease specific differences that have not been published or that I am simply not aware of. However, together with point (1) they question the validity of the classification of these cells as ILCs. As the primary novel observation of this study is that ILC3s may be involved in smoking related asthma, the correct and reliable classification of ILCs is essential. Therefore, I suggest additional work is needed to clarify this.

Other points:

1. fig 2 – The highly significant reduction on blood ILCs in healthy smokers is very striking and somewhat surprising. Has this been reported before? The authors do not explore this observation further, which is a shame as it is perhaps the most striking finding, which, from supplementary fig. 3 suggest that “healthy” smokers essentially lose their circulating ILC2s. Are the authors confident in the observation and can they explore it further, at least in discussion.

2. Supplementary fig 6 - the stats do not look correct here as it is very hard to believe that there is not a significant difference between non-asthmatic smokers and non-smokers in terms of CD4 count as the difference is large and consistent. Together with the increase in asthmatics this appears to

mirror what is observed with ILCs and suggesting that this may not be an ILC specific phenomena.

3. In addition, in this figure, the frequency of CD45RO expressing cell is presented as a frequency of CD4 cells and not of total cells, as it is for ILCs. This is confusing and, given that the total number of CD4 cells in smoking asthmatics is generally higher, it does make you wonder what this figure would look like if it was presented in the same way as for ILCs – that is to say CD45RO+ve CD4 T-cells as a percentage of total cells. Again this hints that the same trend is observed in CD4 T-cells and causes one to question if ILC3s are really the cell type being measured here.

4. Figure 3 e and g, It is very hard to judge if the effect of CSE on CD45RO expression is related to asthma or smoking, as there are only 2 individuals in the non-asthmatic group. The effect in the asthmatic group is convincing

5. Fig. 4a. Some kind of quantification of these images would be helpful. In addition, is this just one experiment? It would be more convincing if it was replicated or a titration effect could be shown.

6. The effect of CSE on epithelial production of IL-1b is convincing. However, as epithelial cells are clearly not present in PBMC this does not provide a mechanism for the effect of CSE on these cells in the PBMC experiments. Perhaps other cells in PBMCs make IL-1 on CSE exposure, exploring this would provide a potential mechanism.

7. Discussion - page 12, I am not sure it is correct to say the ILC3 correlated highly with lung function or numbers of M1 macrophages and PMNs as all R values are below 0.5 and many below 0.4. My understanding is that, as a rule of thumb, an R value of over 0.7 is typically considered to be strong 0.5-0.7 medium and between 0.3 and 0.5 weak. Do the authors have some justification for calling this a strong correlation?

Reviewer #2 (Remarks to the Author):

This study assesses the impact of cigarette smoke on asthma. They showed that:

1. in humans smoking is associated higher frequencies of pro-inflammatory NCR-ILC3s in sputum and memory-like CD45RO+ILC3s in blood.
2. these cells correlate with both the amount of smoking and asthma severity.
3. in vitro exposure of ILCs to CSE induced ILC3 expression of the memory marker CD45RO suggesting the CS can directly activate ILC3s.
4. CSE impaired barrier function of AECs and increased their IL-1b production that is known to activate ILC3s.
5. ILC3s in asthma patients correlated with circulating neutrophils and M1 macs – important in severe asthma – but not circulating eosinophils.

Their conclusion is that cigarette smoking activates ILC3s and increases local and circulating numbers, either directly through compounds in CS or indirectly through altering the lung microenvironment.

These changes are associated with non-allergic and severe asthma.

This is the first assessment of ILC3s in smoking asthma. They link the CS to the development of asthma. However, CS is likely associated with increasing severity in established asthma and this may be more relevant to the current study. The Introduction should discuss this.

This is an important study in the asthma field and adds new insights into the links between ILC3s, smoking and asthma, however there are several issues that need to be addressed

The first sentence of the abstract states that CS causes severe asthma. This needs to be clarified does it cause severe asthma or make asthma more severe?

Do the sputum ILC3s correlate with sputum neutrophils and macrophages?

ILC3 are linked with IL-17 responses in severe asthma and neutrophilic inflammation. What happened with IL-17 in the study. This is important as the authors say that NCR-ILC3 mainly produce IL-17A and F.

Ratios of NCR-/NCR+ILC3s in health controls should be shown in Fig S1.

If there are no differences in ILC3s in the blood where do the ILC3 in the lungs come from in these smokers. Note that CD45RO is increased in both ILC1 and 3 in the blood.

Supplementary figures need to be discussed in sequence in the results eg S3

Fig s6 the CD4+ cells looks significantly decreased in smoking healthy controls. If this is not significant how can other data be throughout the manuscript?

Fig 3c – this is split into current vs former smokers. How does this division affect all of the other results? The former smoker data seems to be skewed by 1 outlier. Fig 3d 0.1% CSE is very low. 1-10% is used in Fig 4. Figs 3e-f need to be discussed in sequence.

Fig 4 uses A549 cells which is a bronchial tumour line. Primary cells would important to assess or at least convention is not to use at least 2 cell lines. TEERs should be measured.

Fig 5 – AHR is a critical readout in asthma please comment.

Fig 6 a and c ILC3s correlate with both sputum and blood neutrophils not just circulating

Fig S7 – are CD11c CD206 established differentiators of M1 and M2 macrophages?

It is surprising that smoking was not associated with elevated circulating neutrophils, since these are short lived cells and smoking is associated with neutrophilic asthma. Can the authors rationalize this?

How can ILC3s correlated with impaired lung function and circulating numbers of M1 macs and neutrophils, how can circulating neutrophils not be associated with impaired lung function?

The clinical data should be described in the results section.

Severe asthma is characterized by a lack responses to steroid therapy – how were CS and ILC3s correlate with treatment responses?

There are some additional clinical and experimental studies of the roles of the impact of CS and chronic obstructive pulmonary disease and ILCs including ILC3s that should be discussed.

Although there are statistically significant differences between ILC1 and 3 in smoking asthma the dot plots seem similar and the significance dependent on a small number of patients. The same could be said for healthy controls. It is normal to validate human studies in a validation cohort – is this possible?

Why is there not an increase in ILC2s in nonsmoking asthma patients? Other studies show that this should occur.

This is quite a specific area – smoking in asthma, is this suitable for Nat Commun?
There are numerous grammatical errors that will need editorial changes.

Minor

Abstract

In the asthma not from the asthma

Intro

Non-allergic asthma is typical more severe rather than can be more severe

Reviewer #3 (Remarks to the Author):

The authors investigated the effects of smoking on ILC composition in asthmatic and healthy donor peripheral blood and sputum samples and found that CD45RO+ "memory-like" ILC3s are increased by smoking only in asthmatic patients but not in healthy individuals.

Major points:

1. The authors should avoid using the term "memory-like ILC3s". Although it was recently demonstrated that activated human ILCs express CD45RO, which has been commonly used to identify memory T cells in humans, there has been no evidence for CD45RO being a memory ILC marker. The authors should perform in vitro experiments to confirm that CD45RO+ ILC3s display memory functions, e.g. more responsive to stimuli and secretion of more cytokines than CD45RO- ILC3s.
2. CD45RO+ and CD45RO- ILC3s should be directly compared by flow cytometry and transcriptome analyses.
3. The study sample size is relatively small and sex distribution is very different between non-smoker and smoker.
4. The cell populations defined by the combination of cKit and ST2 should be further analyzed for the expression of CRTH2, CD161, GATA3, ROR γ t, T-bet and EOMES to confirm their identities and purities.
5. CD45RO expression in sputum ILC3s should be examined, because tissue ILC3s and circulating ILC3s may have very different functions. Also, what is the significance of NCR- ILC3s in smoking asthma patients' sputum?
6. Authors should culture non-smoking asthmatic PB with media collected from CSE treated epithelium culture to see if CSE-induced soluble factors can cause phenotype/function similar to smoking asthmatic patients' ILC3s.

Minor points:

1. What is "induced sputum" ?.
2. Line 113: although authors claim that "smoking in asthma specifically increased the numbers of...", they do not show increased numbers of CD4+CD45RO+ ILC3s, only the frequencies.
3. The order of figures do not follow the order mentioned in the text (ex. Supplementary Figure 5c is mentioned before 5b, Figure 6c is mentioned before 6b, Figures 3g and h are mentioned before e and f).
4. Line 120: what are CD4+ ILC1s?
5. Some control samples only have 2 data points (Figure 3e-h). Should be at least have 3.

Date: December 27th 2021

Manuscript No. NCOMMS-21-07174

Title: Cigarette smoke worsens asthma by inducing memory-like type 3 innate lymphoid cells

Point-by-Point Responses

COMMENTS FROM REVIEWER #1:

This study examines the potential role of ILC3s in smoking induced asthma. The link between ILC2s and allergic asthma has been extensively studied but the potential role of ILC3s is less well established. This study takes several novel approaches, including the quantification of ILC3s in sputum samples, a convenient method of sampling airway cells. Differences in the frequency of certain ILC3 phenotypes between smoking and non-smokers with and without asthma and the (weak) correlations with measures of smoking frequency and disease severity do suggest potentially meaningful differences between these groups have been identified. These findings are novel and of general interest. The paper is very clearly written and the figures are well presented and clear.

However, my underlying concern with this study is whether or not the cells identified are in fact ILCs and not some other cell type. Given that ILCs are generally extremely low frequency, even low level contamination from, say, T-cells expressing low levels of CD3 can have a huge impact. Several other factors add to the uncertainty around whether or not bona fide ILCs are being measured:

RESPONSE: We thank reviewer #1 for reading our manuscript carefully and providing constructive comments. Also, thank you for stating that the current manuscript is “well presented” and took “novel approaches”. Since ILCs are a rare population, we fully understand the concerns raised by the reviewer about T cell contamination. Therefore, we tried to demonstrate that our results are specific to ILC by showing T cells as a control. Moreover, since matured CD4 T cells do not express C-kit (Frumento, G. et al. *Front Immunol*, 2019; Yui, M. A. & Rothenberg, E. V., *Nat Rev Immunol*, 2014), it is unlikely that CD4⁺CD45RO⁺C-kit⁺ ILC3s, which is mainly covered in the current manuscript, are the result of T cell contamination. Although the limitations of the ILC study still exist, we hope that the answer below addresses Reviewer #1's questions.

MAJOR COMMENTS:

1. The frequency of ILCs identified in PBMCs are highly inconsistent with other reports from humans, in which ILC2s are reported as the dominant ILC class, followed by ILC3 precursors and low levels of ILC1s. By contrast supplementary fig.3 suggest

ILC1s are by far the dominant ILC class (median freq approximately 0.175% of total cells, in contrast to less than 0.001% generally reported), followed by ILC3s (here approximately 0.02%, which is consistent) and then extremely low levels of ILC2s (here approximately 0.005%, which is at least an order of magnitude less than expected). As ILC2s are always reported as the most frequent ILC cell type in the blood of healthy controls this does not give any confidence in the identification of ILCs here.

RESPONSE: Thank you for mentioning the critical point. As reviewer 1 noted, in our study, the frequencies of ILC differed from other studies. The reasons for the inconsistency of the ratio of ILC1-3 subset from study to study may be as follows;

(1) Differences in lineage marker; Because lineage markers for ILCs have not yet been established, each group used slightly different markers. We used CD3 ϵ , CD19, CD11b, CD11c, CD14, CD49b, and Fc ϵ R1a as a lineage marker to exclude T, B, NK, monocyte, DC, and granulocytes. However, Carvelli and colleagues used lineage maker containing TCR $\alpha\beta$, TCR $\gamma\delta$, CD94, CD16, CD34, CD123, and CD303, except CD11b, CD11c, CD49b, and showed similar frequencies among ILC1-3 subsets (Carvelli, J. et al., *Front Immunol*, 2019). On the other hand, CD3, CD4, CD11c, CD14, CD19, CD34, CD303, TCR $\alpha\beta$, and TCR $\gamma\delta$ were used by Singh, A. *et al.* and they found the frequency of ILC2s are higher than the other two subsets (Singh, A. et al., *Cell Rep*, 2020). Therefore, the different usage of lineage markers would be the one reason why ILC frequencies and the ratio of each ILC subsets were vary among studies.

(2) Differences between cohorts and human subjects; Even when the same lineage markers are used, the proportion of ILCs varies from person to person. Recently, Bonne-Année *et al.* showed that the distribution of ILC subsets is not consistent even in the peripheral blood from healthy controls, with ILC1s being the predominant population (CD4, CD8 added for lineage marker, Bonne-Annee, S. et al., *Sci Rep*, 2019).

[REDACTED]

Also, Vely, F. *et al.* showed that the frequencies of ILC1-3 subsets from adult peripheral blood were similar (TCR $\alpha\beta$, and TCR $\gamma\delta$ added for lineage marker, Vely, F. *et al.*, *Nat Immunol*, 2016). Therefore, various factors, such as differences in the cohorts and/or lineage markers, influence inconsistent observations of ILC frequency between studies.

[REDACTED]

2. The identification of CD45RO⁺ ILC3s in the blood. The authors justify this with reference to a paper by Shikhagaieer *et al.* showing elevated CD45RO expressing ILC3s in blood of individuals with COPD. However, I cannot find any evidence of this data in the reference given. The authors do observe expression of CD45RO on NRP1⁺ ILC3s in lung tissue and in tonsils, but I find no data on CD45RO expression on blood ILC3s. The study does, however, present clear evidence that blood ILC3 do not express NRP1, and in tissue CD45RO is only expressed on NRP1⁺ve ILC3s. Therefore, I do not understand the rationale for using this marker, especially when other studies (doi: 10.4049/jimmunol.1501491) suggests that CD45RO is expressed on ILC1s in human PBMC but not on ILC2s and ILC3s.

RESPONSE: Thanks for finding our error. We changed the descriptions about the study by Shikhagaie *et al.* on Page 6, Line 104, as follows:

“This choice of markers reflects the fact that (i) patients with chronic obstructive pulmonary disease (COPD) have higher peripheral blood frequencies of ILC3s that express CD45RO”

→ “These cells are elevated in the lung and tonsils of patients with chronic obstructive pulmonary disease (COPD).”

The reason for choosing the CD45RA/CD45RO marker was to confirm that the ILCs could also express the naive/memory marker. Similar to our question, Van *et al.* showed that CD45RO expressing ILC2s are increased in the blood and nasal polyp of CRS patients (van der Ploeg, E. K. *et al.* *Sci Immunol*, 2021). Together with our current results, these results suggest that ILCs expressing memory markers (CD45RO) can be distinguished from naive ILCs.

[REDACTED]

In the study of Roan *et al.* mentioned by reviewer #1, CD45RO expression was highest in ILC1 (92.4-7.8%), but ILC2 and 3 also expressed CD45RO (28.1-1.3% of ILC2s, and 12.7-0.9% of ILC3s) in PBMCs from healthy control (Roan, F. et al., *J Immunol*, 2016). We also found that ILC3 barely expresses CD45RO in healthy conditions, but CD45RO expression increased in asthmatics. Therefore, we speculated that repeated challenges by smoking could induce the expression of CD45RO in ILC3s.

3. Expression of CD4 on circulating human ILC3s. In addition to the apparently unusual expression of CD45RO on circulating ILC3s, the biggest differences are observed when these cells are divided into CD4⁺ and CD4⁻ ILC3s. However, I am not aware of other publications that have identified CD4 expression on circulating human ILC3s. The following publication, for example, identifies potential expression on ILC1-like cells but finds no expression on ILC3s (doi: 10.4049/jimmunol.1501491).

RESPONSE: Although it is a mouse study, it has also been reported that retinoic acid induces ROR γ t expression in NCR-CD4⁻ILC3 precursors and then ROR γ t differentiates these precursors into NCR-CD4⁺ ILC3 (van de Pavert et al., *Int Immunol* 2016). Assuming that the CD4-expressing NCR⁻ILC3s would be in a more differentiated form, we further confirmed CD4 expression, and found that CD4⁺CD45RO⁺ILC3 was significantly increased under smoking conditions.

The results of Roan *et al.* were from healthy controls. When comparing CD4 expressing ILC1s and ILC3s from healthy controls (HC), non-smoking asthmatics (Non-smoker), and smoking asthmatics (Smoker), CD4⁺ILC1s were uniformly present in all conditions, whereas CD4⁺ILC3s were specifically increased in smoking asthma patients. Although the current study has limitations in not ascertaining whether blood CD4⁺ILC3s are originated from lung tissue, it does suggest that smoking conditions may cause an increase in circulating CD4-expressing ILC3s.

MINOR COMMENTS:

1. fig 2 – The highly significant reduction on blood ILCs in healthy smokers is very

striking and somewhat surprising. Has this been reported before? The authors do not explore these observations further, which is a shame as it is perhaps the most striking finding, which, from supplementary fig. 3 suggest that “healthy” smokers essentially lose their circulating ILC2s. Are the authors confident in the observation and can they explore it further, at least in discussion.

RESPONSE: As the reviewers pointed out, our results showed a significant reduction of ILCs and CD4 T cells in healthy smokers. Several papers already suggest the immunosuppressive effect of smoking; 1) Nicotine, the main component of cigarette smoke, inhibits antigen-specific T cell generations by reducing antigen uptake of dendritic cells (Nouri-Shirazi, M. & Guinet, E., *Immunology*, 2003). 2) cigarette smoke downregulates IL-33 receptor expression in pulmonary ILCs, suppressing effector function (Kearley, J. et al., *Immunity*, 2015). It has also been reported that the nicotine receptor, $\alpha 7nAChR$, is highly expressed in ILC2s, and this signaling attenuates the function of ILC2s (Galle-Treger, L. et al., *Nat Commun*, 2016). Therefore, differences in the effects of smoking between asthmatics and healthy controls suggest that the effects of cigarette smoke may be determined by various factors. We have added a discussion about the effects of smoking on immune cells (Page 15 Lines 289 – 292).

2. Supplementary fig 6 - the stats do not look correct here as it is very hard to believe that there is not a significant difference between non-asthmatic smokers and non-smokers in terms of CD4 count as the difference is large and consistent. Together with the increase in asthmatics this appears to mirror what is observed with ILCs and suggesting that this may not be an ILC specific phenomena.

RESPONSE: Thanks for finding the mistake. As the reviewer pointed out, the frequency of CD4⁺ T cells was significantly decreased in the healthy smoking group ($p < 0.0001$). We corrected Fig. S6b as blow.

3. In addition, in this figure, the frequency of CD45RO expressing cell is presented as a frequency of CD4 cells and not of total cells, as it is for ILCs. This is confusing and, given that the total number of CD4 cells in smoking asthmatics is generally higher, it does make you wonder what this figure would look like if it was presented in the same way as for ILCs – that is to say CD45RO⁺ve CD4 T-cells as a percentage of total cells. Again, this hints that the same trend is observed in CD4 T-cells and causes one

to question if ILC3s are really the cell type being measured here.

RESPONSE: As suggested by the reviewer, we replaced the graph of the frequencies of CD45RO⁺ CD4⁺T cells as % in total cells like ILC3s (Fig. S6d). There was no significant difference in CD4 T cells between smoking and non-smoker groups even when compared by % of the total number.

4. Figure 3 e and g, It is very hard to judge if the effect of CSE on CD45RO expression is related to asthma or smoking, as there are only 2 individuals in the non-asthmatic group. The effect in the asthmatic group is convincing

RESPONSE: We added more samples (n = 10) and re-analyzed the results in Fig. 3e-h. Although we couldn't confirm the decrease in CD45RA expressions from ILC3s by CSE treatment, an increase in CD45RO expression was concordantly observed (Fig 3e-h).

5. Fig. 4a. Some kind of quantification of these images would be helpful. In addition, is this just one experiment? It would be more convincing if it was replicated or a titration effect could be shown.

RESPONSE: Thanks for the constructive advice. The Images in Fig. 4a are the representative data from three different experiments. We added a graph of E-Cadherin quantification of IF imaging to Fig 4a. We also analyzed the changes of transepithelial electrical resistance (TEER) and found TEER was significantly decreased in CSE-treated A549 cells, confirming that CSE treatment impairs the integrity of the epithelial barrier. In addition to A549, we examined various epithelial cell lines (RPMI2650, BEAS-2B, and MLE12) and added these results to Fig 4b.

Figure 4 - added more cell lines, quantified data, and TEER results

a – added the quantified graph

b – TEER results of cell lines

6. The effect of CSE on epithelial production of IL-1b is convincing. However, as epithelial cells are clearly not present in PBMC this does not provide a mechanism for the effect of CSE on these cells in the PBMC experiments. Perhaps other cells in PBMCs make IL-1b on CSE exposure, exploring this would provide a potential mechanism.

RESPONSE: Thanks for the good point. To elucidate the cellular source of IL-1 β in PBMCs by smoking, we analyzed IL-1 β producing cells from healthy controls and asthmatic patients, and added these data in Supplementary Fig. 8. IL-1 β expressing cells were increased in asthma patients, but there was no difference between non-smoker and smoker groups (Supplementary Fig. 8a). Among the various cell types in PBMCs, DCs and classical monocytes were the predominant cellular source of IL-1 β , but IL-1 β production was not increased in smoking groups (Supplementary Fig. 8b-c). These results suggest that the contribution of blood immune cells to the increase of ILC3s is not significant.

ILCs have long been considered tissue-resident cells, but several recent studies have suggested that ILCs egress from tissue under inflammatory conditions (Willinger, T., *Front Immunol*, 2019; Kim, M. H. et al., *Immunity*, 2015; Kastele, V. et al., *Mucosal immunology*, 2021). Although further studies are needed, cigarette smoking may damage the epithelial cells, allowing airway ILC3s to migrate into the bloodstream.

7. Discussion - page 12, I am not sure it is correct to as say the ILC3 correlated highly with lung function or numbers of M1 macrophages and PMNs as all R values are

below 0.5 and many below 0.4. My understanding is that, as a rule of thumb, an R value of over 0.7 is typically considered to be strong 0.5–0.7 medium and between 0.3 and 0.5 weak. Do the authors have some justification for calling this a strong correlation?

RESPONSE: We used the word “highly” to emphasize the significance of correlation. However, we understand the reviewer’s point that this word could lead to misinterpretation, so we have changed that description on Page 15 Line 267, as follows:

“Given that the elevated ILC3 numbers in smokers with asthma correlated highly with not only decreased lung function but also M1 macrophages”

→ “Given that the elevated ILC3 frequencies in smokers with asthma correlated positively with not only with smoking amount and decreased lung function but also M1 macrophage and circulating neutrophil frequencies,”

The interpretation of the correlation coefficient (R-value) varies according to research fields. In medical studies using patient samples, an R-value greater than 0.3 is considered fair (Akoglu, H., *Turk J Emerg Med*, 2018). Most of the R-values in our study are above 0.3, and a higher correlation is expected if the number of samples is sufficient.

[REDACTED]

COMMENTS FROM REVIEWER #2:

This study assesses the impact of cigarette smoke on asthma. They showed that:

1. in humans smoking is associated higher frequencies of pro-inflammatory NCR-ILC3s in sputum and memory-like CD45RO+ILC3s in blood.
2. these cells correlate with both the amount of smoking and asthma severity.
3. in vitro exposure of ILCs to CSE induced ILC3 expression of the memory marker CD45RO suggesting the CS can directly activate ILC3s.
4. CSE impaired barrier function of AECs and increased their IL-1b production that is known to activate ILC3s.
5. ILC3s in asthma patients correlated with circulating neuts and M1 macs – important in severe asthma – but not circulating eosinophils.

Their conclusion is that cigarette smoking activates ILC3s and increases local and circulating numbers, either directly through compounds in CS or indirectly through

altering the lung microenvironment. These changes are associated with non-allergic and severe asthma.

This is the first assessment of ILC3s in smoking asthma. They link the CS to the development of asthma. However, CS is likely associated with increasing severity in established asthma and this may be more relevant to the current study. The Introduction should discuss this.

This is an important study in the asthma field and adds new insights into the links between ILC3s, smoking and asthma, however there are several issues that need to be addressed

RESPONSE: We would like to express our deep gratitude to the reviewer #2 who regarded this study as an important and new insight into the field of asthma. Although smoking-induced asthma exacerbations were not evident in the current asthma cohort, FEV₁, FEV₁/FVC, and ACT levels were significantly lower in smoking asthmatics.

MAJOR COMMENTS:

1. The first sentence of the abstract states that CS causes severe asthma. This needs to be clarified does it cause severe asthma or make asthma more severe?

RESPONSE: We apologize for the confusing description. Smoking is involved in both the development and exacerbation of asthma, but we mean that smoking makes asthma worse. Therefore, we changed the first sentence of the abstract on Page 3 Line 2, as follows:

“Although cigarette smoking causes severe asthma,”

→ “Although cigarette smoking can exacerbate asthma,”

2. Do the sputum ILC3s correlate with sputum neutrophils and macrophages?

RESPONSE: Due to the limitation in the number of sputum cells, we couldn't check the sputum neutrophils in this study. Instead, we stained sputum macrophages and presented a positive correlation between ILC3s and sputum M1 macrophages in Fig. 6e.

3. ILC3 are linked with IL-17 responses in severe asthma and neutrophilic inflammation. What happened with IL-17 in the study. This is important as the authors say that NCR-ILC3 mainly produce IL-17A and F.

RESPONSE: Thanks for raising an important issue. As the reviewer's comment, IL-17 plays an essential role in severe and neutrophilic asthma. Unfortunately, immune cells obtained from induced sputum undergo apoptosis when stimulated for cytokine analysis. Therefore, we analyzed IL-17A expression in blood ILCs, which are NCR-ILC3s. Since we suggest that an increase in CD4+CD45RO+ILCs contributes to the exacerbation of smoking asthma, we compared the IL-17A expression between

CD45RA+, CD4-CD45RO+, and CD4+CD45RO+ populations. The CD4+CD45RO+ ILCs are the major producer of IL-17A, as shown in the figure below, and we added this result to Fig 2h.

4. Ratios of NCR⁻/NCR⁺ ILC3s in health controls should be shown in Fig S1.

RESPONSE: We added the graph of ratios of NCR⁻/NCR⁺ ILC3s in healthy controls in Fig. S1b. There was no difference in the ratios of NCR⁻/NCR⁺ ILC3s between non-smokers and smokers

5. If there are no differences in ILC3s in the blood where do the ILC3 in the lungs come from in these smokers. Note that CD45RO is increased in both ILC1 and 3 in the blood.

RESPONSE: Although the proliferation of ILC3 cannot be confirmed due to the limitation of the sputum sample, the increase in ILC3s in the sputum may be a result of the local proliferation of ILC3s. Although immune cell infiltration from the blood into inflamed tissues, called trans-endothelial cell migration (TEM), is well known phenomenon, recent studies have also reported reverse endothelial cell migration (rTEM), in which leukocytes leave the site of inflammation and enter the circulation (Burn T et al., *Cell Mol Life Sci*, 2017; Hirano Y et al., *Biol Chem*, 2016).

Regarding CD45RO expression, there was a trend toward increased CD45RO expression in ILC3s and ILC1s. However, when we added more samples for revision, there was no statistical significance in the expression of CD45RO in ILC1s and ILC3s (Fig 2e). When CD45RO-expressing cells were divided according to CD4 expression, only CD4+CD45RO+ ILC3s were significantly increased and produced IL-17A (figure on response 3). Furthermore, *in vitro* CSE treatment increased the expression of CD45RO only in ILC3s (Fig 3g). For these reasons, we focused on ILC3s rather than ILC1s in the current manuscript.

6. Supplementary figures need to be discussed in sequence in the results eg S3
RESPONSE: Thank you for your careful review. We checked the manuscript and rearranged the figures in the order of presentation.

7. Fig s6 the CD4⁺ cells looks significantly decreased in smoking healthy controls. If this is not significant how can other data be throughout the manuscript?

RESPONSE: Thanks for finding the mistake. As the reviewer pointed out, the frequency of CD4⁺ T cells was significantly decreased in the healthy smoking group ($p < 0.0001$). We corrected Fig. S6b as blow.

8. Fig 3c – this is split into current vs former smokers. How does this division affect all of the other results? The former smoker data seems to be skewed by 1 outlier. Fig 3d 0.1% CSE is very low. 1-10% is used in Fig 4. Figs 3e-f need to be discussed in sequence.

RESPONSE: When we excluded the outlier in the former smokers, the frequency of CD4⁺CD45RO⁺ILC3s was still significantly increased in former smokers compared to non-smokers ($p < 0.05$). However, the frequency of CD4⁺CD45RO⁺ILC1s was not changed according to former/current smokers, as shown below.

The difference in CSE concentration in Fig. 3d-h and Fig. 4 is because the cells used in each experiment are different. In Fig. 3d-h, we used PBMC, and live cells started to decrease when we treated 1% CSE and almost disappeared at 5% CSE treatment.

Frequency of live cells in CSE-treated PBMCs according to CSE concentration.

In Figure 4, we used immortalized cell lines; therefore, a higher concentration of CSE did not affect cell viability. Also, when we titrated the concentration of CSE, the difference between gene expression and the barrier integrity was most evident at 10% CSE treatment. As the reviewer pointed out, it would be best to observe the same effect at the same concentration, but please understand that we have to use different concentrations of CSE depending on the cell type used.

9. Fig 4 uses A549 cells which is a bronchial tumour line. Primary cells would important to assess or at least convention is not to use at least 2 cell lines. TEERs should be measured.

RESPONSE: Thanks for the critical point. Unfortunately, it takes too much time to purchase a primary epithelial cell line due to COVID researches. Instead, we additionally tested RPMI2650 (human nasal epithelial cells), BEAS-2B cells (human bronchial epithelial cell line), and MLE12 cells (mouse lung epithelial cell line). Similar to A549 cell lines, *IL1B* gene expression was increased after CSE treatment in three epithelial cell lines. Also, TEER was significantly decreased in all cell lines after CSE treatment, which suggested that CSE disrupts epithelial barrier integrity. Now, we added these results in Fig. 4.

Figure 4 - added more cell lines, quantified data, and TEER results

a – added the quantified graph

b – TEER results of cell lines

c

d

10. Fig 5 – AHR is a critical readout in asthma please comment.

RESPONSE: As pointed out by reviewers, we added the PC₂₀ level (provocative concentration of methacholine required to decrease FEV₁ by 20%) in Table 1. There was no significant change in the PC₂₀ levels of non-smokers and smokers in asthma patients (P-value = 0.1364; Table 1). However, FEV₁ level, FEV₁/FVC ratio, and ACT level of smokers were significantly lower, and ACQ level was significantly higher than that of non-smokers in patients with asthma. Since PC₂₀ levels did not differ between non-smokers and smokers in patients with asthma, we could not analyze the correlation between immune cells and PC₂₀ levels in Fig. 5 and Fig. S10 (previously Fig. S8).

Table 1. Characteristics of the study subjects

	Healthy Control		Asthma		P value
	Non-smoker	Smoker	Non-smoker	Smoker	
n	13	11	33	58	
Age (Yr)	55.1±5.7	59.0±13.3	52.1±14.0	55.1±11.4	
Sex (M/F)	6/7	8/3	6/27	49/9	
Body mass index (kg/m ²)	23.0±2.6	26.3±2.7	23.6±3.1	25.8±4.9	
Allergic rhinitis, n (%)	0 (0)	0 (0)	21 (63.6)	32 (55.2)	
Atopic dermatitis, n (%)	0 (0)	0 (0)	3 (9.1)	6 (10.4)	
FVC (mL)	3465.0±837.0	4002.0±732.4	3180.0±1014.0	3646.0±878.3	0.1037
FVC (%)	107.3±12.6	100.7±19.1	103.5±13.8	88.2±15.1	<0.0001
FEV ₁ (mL)	2749.0±639.6	3177.0±595.3	2542.0±657.5	2444.0±786.6	0.0747
FEV ₁ (%)	103.5±12.6	104.8±14.1	95.6±10.5	76.1±16.7	<0.0001
FEV ₁ /FVC (%)	79.8±6.4	79.6±7.7	77.0±5.4	65.9±11.2	<0.0001
ACQ	0	0.1±0.1	5.8±4.4	7.0±5.3	0.0271
ACT	25	25	21.8±3.1	20.2±3.4	0.0061
PC ₂₀ (mg/ml)	n.d	n.d	10.9±9.0	5.5±6.2	0.1364
OCS, n (%)	n.d	n.d	5 (15.2)	13 (22.4)	
Hemoglobin (g/dL)	n.d	n.d	13.3±2.4	15.0±1.7	0.0003
WBC (/μL)	n.d	n.d	7055.0±2830.0	7948.0±3650.0	0.2978
Lymphocytes (/μL)	n.d	n.d	1958.0±740.0	2317.0±1310.0	0.2548
Monocytes (/μL)	n.d	n.d	428.3±153.1	576.8±365.5	0.1301
Eosinophils (/μL)	n.d	n.d	317.3±349.3	318.6±314.5	0.6822
Neutrophils (/μL)	n.d	n.d	4174.0±2510.0	4655.0±3514.0	0.6754
Basophils (/μL)	n.d	n.d	39.7±23.4	39.0±22.2	0.6179

FVC: forced vital capacity; FEV₁: Forced expiratory volume in one second; ACQ: asthma control questionnaire; ACT: asthma control test; PC₂₀: provocative concentration of methacholine required to decrease FEV₁ by 20%; OCS: oral corticosteroid; WBC: white blood cell

11. Fig 6 a and c ILC3s correlate with both sputum and blood neutrophils not just circulating

RESPONSE: We could not analyze neutrophils in sputum because of the limited number of cells available in the sputum sample, as answered in a previous comment. Although sputum neutrophil counts are a prominent indicator of asthma severity, circulating neutrophil counts are also associated with lung function and non-eosinophilic asthma characteristics (Lewis SA et al., *Chest*, 2001; Nadif R et al., *Thorax*, 2009). Therefore, it would be best to analyze the correlation between sputum neutrophils and ILC3s, but as an alternative, the correlation between sputum ILC3s and blood neutrophils were analyzed.

12. Fig S7 – are CD11c CD206 established differentiators of M1 and M2 macrophages?

RESPONSE: Various markers are used to confirm the M1/M2 differentiation of macrophages. It would have been nice if more diverse markers were used, but the M1 marker, CD11c, and the M2 marker, CD206, are among the commonly used surface markers (Jayasingam et al. *Front Oncol.*, 2020).

[REDACTED]

13. It is surprising that smoking was not associated with elevated circulating neutrophils, since these are short lived cells and smoking is associated with neutrophilic asthma. Can the authors rationalize this?

RESPONSE: When asthma patients were divided into non-smokers and smokers, blood neutrophils did not differ between groups, as shown in Fig. S9a (previously Fig. S7a).

The number of neutrophils may be increased in smoking asthma compared to normal subjects, but the difference between smoking and non-smoking asthmatics was not significant. However, a positive correlation was observed between smoking amounts (Pack-Year) and the number of neutrophils in the blood. Now, we added this result in Fig. S9b. It would also be more meaningful to compare the activation index of neutrophils (such as MPO, ROS, NETs) rather than numerical comparisons.

14. How can ILC3s correlated with impaired lung function and circulating numbers of M1 macs and neutrophils, how can circulating neutrophils not be associated with impaired lung function?

RESPONSE: Previous results showed a negative correlation between circulating neutrophils and lung function but failed to reach statistical significance. We added patient samples during the revision process and found a significant correlation between neutrophils and impaired lung function as shown below. Therefore, we edited the description of Fig. S10 on Page 12 Line 235-237, as follows:

“However, despite the fact that ILC3 frequencies correlated positively with M1 macrophage and circulating neutrophil numbers, neither M1 macrophages nor neutrophils associated with impaired lung function (Supplementary Fig. 8a-d). This suggests they may play a more downstream role in asthma pathogenesis than ILC3s.”

→ “Notably, sputum M1 macrophage frequencies did not associate with impaired lung function, unlike neutrophil counts (Supplementary Fig. 10a–d). This suggests that neutrophils interact more closely with ILC3 subsets in the pathogenesis of smoking asthma than M1 macrophages.”

15. The clinical data should be described in the results section.

RESPONSE: According to the reviewer's suggestion, we added the description of clinical data in the result section (Page 5 Line 66–70), as follows:

“Compared to the non-smokers with asthma, the smokers with asthma had significantly impaired lung function, as shown by (i) lower forced expiration volume of 1 second % (FEV₁ %), ratio of FEV₁/FVC, and asthma control test (ACT) scores, and (ii) higher asthma control questionnaire (ACQ) scores. As indicated by the PC20 value, the two groups did not differ in airway hyperresponsiveness. The healthy non-smokers and smokers did not differ significantly in terms of any lung function indicator (Table 1)”

16. Severe asthma is characterized by a lack responses to steroid therapy – how were CS and ILC3s correlate with treatment responses?

RESPONSE: Thanks for raising an important issue. There was no difference in oral corticosteroid (OCS) use in smoking and non-smoker asthma. In addition, the ILC3 frequency was not affected by the presence or absence of OCS and the dose. We added the number of patients who are using OCS in Table 1.

17. There are some additional clinical and experimental studies of the roles of the impact of CS and chronic obstructive pulmonary disease and ILCs including ILC3s that should be discussed.

RESPONSE: In response to the reviewer's suggestion, we added other references indicating an increase in ILC3s in cigarette smoke exposure models and COPD, and added them to the discussion section on Page 13 Line 253-256, as follows:

"These findings are supported by several studies that suggest ILC3s are linked to smoking. First, COPD patients have increased NKp44-ILC3 frequencies in their blood compared to control groups (Bal et al. *Nat. Immunol.*, 2016; De Grove et al., *PLoS One*, 2016). Second, the lungs of cigarette smoke-exposed mice have elevated IL-17A- or IL-22-expressing ILC3 frequencies (Starkey et al., *Eur. Respir. J.* 2019)."

18. Although there are statistically significant differences between ILC1 and 3 in smoking asthma, the dot plots seem similar and the significance dependent on a small number of patients. The same could be said for healthy controls. It is normal to validate human studies in a validation cohort – is this possible?

RESPONSE: Please understand that we couldn't validate this study in additional cohorts. Instead, we recruited more asthmatics as well as healthy controls (healthy smoker (1), non-smoking asthma patients (9), and smoking asthma patients (18)). The addition of the sample resulted in a more significant increase in ILC3s.

19. Why is there not an increase in ILC2s in nonsmoking asthma patients? Other studies show that this should occur.

RESPONSE: The Graphs in Fig. 1 and 2 are the comparison of ILC2s between non-smokers and smokers in patients with asthma. When we compared the frequency of ILC2s between healthy controls and asthma patients, ILC2s were significantly increased in both induced sputum and PMBC in asthmatic patients, as shown in the figure.

Comparison of ILC2s between healthy controls and patients with asthma.

20. This is quite a specific area – smoking in asthma, is this suitable for Nat Commun?

RESPONSE: It is stated on the website that the purpose and scope of Nature Communications are to publish research that represents important advances for professionals in the respective fields of biology, health, physics, chemistry, and earth sciences. We think the current study is suitable for the health and biology scope and valuable as a study providing new insights into the role of memory-like ILC3 in smoking-induced asthma exacerbations.

21. There are numerous grammatical errors that will need editorial changes.

RESPONSE: This manuscript has been re-edited by a Ph.D. holder in Immunology. The certification of editing is attached.

[REDACTED]

MINOR COMMENTS:

1. Abstract - In the asthma not from the asthma

RESPONSE: The abstract has been revised as the reviewer's comment (Page 2, line 12).

2. Intro - Non-allergic asthma is typical more severe rather than can be more severe

RESPONSE: We corrected the manuscript as the reviewer's comment (Page 3 Line 25).

COMMENTS FROM REVIEWER #3:

The authors investigated the effects of smoking on ILC composition in asthmatic and healthy donor peripheral blood and sputum samples and found that CD45RO+ "memory-like" ILC3s are increased by smoking only in asthmatic patients but not in healthy individuals.

RESPONSE: We appreciate reviewer #3 for his/her valuable comments.

MAJOR COMMENTS:

1. The authors should avoid using the term "memory-like ILC3s". Although it was recently demonstrated that activated human ILCs express CD45RO, which has been commonly used to identify memory T cells in humans, there has been no evidence for CD45RO being a memory ILC marker. The authors should perform in vitro experiments to confirm that CD45RO+ ILC3s display memory functions, e.g., more responsive to stimuli and secretion of more cytokines than CD45RO- ILC3s.

RESPONSE: Because CD45RO is a common marker of memory T cells, we fully understand the reviewer's concerns that using CD45RO as a memory marker in ILCs is not appropriate. However, recent studies, including ours, suggest that ILCs may also exhibit memory features; i) Martinez-Gonzalez *et al.* showed that allergen-experienced ILC2s acquire antigen non-specific memory-like properties, and enhance type 2 lung inflammation (Martinez-Gonzalez et al., *Immunity*, 2016). ii) Verma *et al.* demonstrated a distinct epigenetic landscape of memory ILC2s by ATAC-sequencing (Mukesh Verma et al., *J Exp Med*, 2021). These studies described the memory-like properties of ILC2s in asthmatic disease. In addition, a very recent study identified that human CD45RO-expressing ILC2s were derived from resting CD45RA+ ILC2s by epithelial alarmins such as IL-33 and TSLP (E. K. van der Ploeg et al. *Sci Immunol*, 2021).

Whether ILC3s expressing CD45RO are a real "memory" ILC3s requires more research, but we would like to retain the term "memory-like" ILC3 for the following reasons. The frequency of CD45RO⁺ILC3s remains high in former smokers as well as current smokers (Fig. 3c). Second, when comparing IL-17A ILCs according to CD45RO expression, CD45RO⁺ILCs (especially CD4⁺CD45RO⁺ILCs) express a significantly higher amount of IL-17A. Therefore, CD45RO expressing ILCs are more sensitive and actively respond to stimuli like memory cells.

2. CD45RO⁺ and CD45RO⁻ ILC3s should be directly compared by flow cytometry and transcriptome analyses.

RESPONSE: As we answered in response 1, CD45RO⁺ILCs produce higher IL-17A than CD45RA, and CD4⁺CD45RO⁺ILCs expressed IL-17A at the highest level. Now, we added the result in Fig. 2h.

3. The study sample size is relatively small and sex distribution is very different between non-smoker and smoker.

RESPONSE: We agree with the reviewer's comment. Although we tried to recruit male and female patients equally, gender matching was impossible because male smokers outnumber female smokers (35.7% of males and 6.7% of females are smoking in South Korea, 2019; KCDA, Korea National Health and Nutrition Examination Survey). However, we compensated for the small sample size by recruiting additional 28 patients and obtained similar results.

4. The cell populations defined by the combination of cKit and ST2 should be further analyzed for the expression of CRTH2, CD161, GATA3, RORγt, T-bet and EOMES to confirm their identities and purities.

RESPONSE: Thanks for raising a critical issue. We examined the signature transcriptional factors in each subset of ILCs; ILC1s (C-kit⁻ST2⁻), ILC2s (ST2⁺), and ILC3s (C-kit⁺ST2⁻). As expected, T-bet in ILC1s, GATA3 in ILC2s, and RORγt in ILC3s were highly expressed, suggesting that currently used markers can divide ILC subsets well

5. CD45RO expression in sputum ILC3s should be examined, because tissue ILC3s and circulating ILC3s may have very different functions. Also, what is the significance of NCR- ILC3s in smoking asthma patients' sputum?

RESPONSE: As suggested by the reviewer, it would be necessary to identify CD45RO expression in sputum ILC3s. However, the number of immune cells that can be obtained from sputum is very small, and the frequency of ILC is much lower. Therefore, please understand that the analysis of CD45RO expression in sputum ILC3s is inevitably inaccurate and currently not possible.

NCR⁺ILC3s secrete IL-22 to maintain epithelial barrier integrity whereas NCR⁻ILC3s mainly produce IL-17A and IL-17F, which are known to induce pathogenic inflammation (Kerim Hoorweg et al., *Front. Immunol.*, 2012; Naoko Satoh-Takayama et al., *Immunity*, 2008; Jenny Mjösberg et al., *J. Allergy Clin. Immunol.*, 2016). Therefore, elevated NCR-ILC3 in the sputum suggests their potential to exacerbate neutrophil inflammation *via* IL-17A production.

6. Authors should culture non-smoking asthmatic PB with media collected from CSE treated epithelium culture to see if CSE-induced soluble factors can cause phenotype/function similar to smoking asthmatic patients' ILC3s.

RESPONSE: To elucidate the effects of soluble factors from CSE-treated epithelial cells, we cultured PBMCs with CSE-treated A549 cell culture supernatant. No significant changes in ILC frequency and CD45RA/RO expression were observed when we co-cultured PBMC with CSE-treated A549 culture supernatants, although we have tried several culture conditions. Such unexpected results may occur for the following reasons: i) The concentration of soluble factors in CSE-treated A549 cells was low, so it failed to induce changes in the ILC phenotype or function in PBMCs. ii) The soluble factor secreted from epithelial cells requires further processing such as proteolysis, and *in vitro* culture does not meet these conditions.

MINOR COMMENTS:

1. What is “induced sputum” ?

RESPONSE: Induced sputum is sputum that induced by inhaling hypertonic saline. Inhaled saline irritates airway epithelium and induces mucus production (Paggiaro, P. L. et al., *Eur Respir J Suppl*, 2002). Induce sputum, which contains airway infiltrated immune cells, cytokines, and antimicrobial peptides have been used as a non-invasive sampling technique to represent the lung immunological environment (Bhowmik, A. et al., *Thorax*, 1998; Tangedal, S. et al., *PLoS One*, 2019; Henig, N. R. et al., *Thorax*, 2001; Angelica Tiotiu et al., *Allergy*, 2020).

2. Line 113: although authors claim that “smoking in asthma specifically increased the numbers of...,” they do not show increased numbers of CD4+CD45RO+ ILC3s, only the frequencies.

RESPONSE: Thank you for letting us know about our mistake. We changed “the numbers of” into “the frequencies of” (Page 7 Line 120 and others throughout the manuscript).

3. The order of figures do not follow the order mentioned in the text (ex. Supplementary Figure 5c is mentioned before 5b, Figure 6c is mentioned before 6b, Figures 3g and h are mentioned before e and f).

RESPONSE: We have rearranged the all figures and descriptions in sequence throughout the manuscript.

4. Line 120: what are CD4+ ILC1s?

RESPONSE: Roan *et al.* analyzed blood ILC markers and identified CD4+ILC1s. They

suggested that CD4+ILC1s could be more activated in inflammatory conditions by highly expressing chemokine receptors and IL-6R (Table from Roan, F. et al, *J Immunol*, 2016).

[REDACTED]

5. Some control samples only have 2 data points (Figure 3e-h). Should be at least have 3.

RESPONSE: We added more samples (n = 10) and re-analyzed the results in Fig. 3e-h. Although we couldn't confirm the decrease in CD45RA expressions from ILC3s by CSE treatment, an increase in CD45RO expressions was observed (Fig 3e-h).

REVIEWER COMMENTS

Reviewer #1 (Remarks to the Author):

The authors have done an excellent job of considering and responding to my comments and queries where possible. They have provided excellent references to back up some of the claims I was unsure about and included useful new data to strengthen their conclusions. The manuscript has been significantly improved, and I have no further reservations. I recommend this paper for publication without other changes.

Reviewer #3 (Remarks to the Author):

The authors responded to my criticism and revised the manuscript accordingly. The revised manuscript is acceptable for publication.

Mediating comments to authors' response to missing Reviewer #3:

(Uploaded by editor with R#2's permission):

I think the authors properly addressed most of the comments from Reviewer 2 with one exception. The reviewer wanted analyses of cells in induced sputum as they likely reflect inflammatory cells in the lung tissue. In response, the authors stated that they could not do the analyses due to very low cell numbers in induced sputum and they analyzed peripheral blood cells instead. It is puzzling that why the number of cells in sputum is so low. In fact, in the revised manuscript, they show analyses of ILC2s in sputum, indicating that the authors could recover sufficient cells in sputum to analyze for ILC2s. The authors also state that they could not analyze for IL-17 producing ILC3s in sputum as they die as they are stimulated in vitro. It suggests that there were enough ILC3s in sputum to be detected. The authors should clarify why they are unable to present the data on ILC3s, neutrophils and macrophage in sputum.

Date: February 28th 2022

Manuscript No. NCOMMS-21-07174

Title: Cigarette smoke worsens asthma by inducing memory-like type 3 innate lymphoid cells

Point-by-Point Responses

Reviewer #1 (Remarks to the Author):

COMMENT: The authors have done an excellent job of considering and responding to my comments and queries where possible. They have provided excellent references to back up some of the claims I was unsure about and included useful new data to strengthen their conclusions. The manuscript has been significantly improved, and I have no further reservations. I recommend this paper for publication without other changes.

RESPONSE: We thank reviewer #1 for his/her time and effort in evaluating this revision and for recommending that this manuscript be published.

Reviewer #2 (Uploaded by editor with R#2's permission):

COMMENT: I think the authors properly addressed most of the comments from Reviewer 2 with one exception. The reviewer wanted analyses of cells in induced sputum as they likely reflect inflammatory cells in the lung tissue. In response, the authors stated that they could not do the analyses due to very low cell numbers in induced sputum and they analyzed peripheral blood cells instead. It is puzzling that why the number of cells in sputum is so low. In fact, in the revised manuscript, they show analyses of ILC2s in sputum, indicating that the authors could recover sufficient cells in sputum to analyze for ILC2s. The authors also state that they could not analyze for IL-17 producing ILC3s in sputum as they die as they are stimulated in vitro. It suggests that there were enough ILC3s in sputum to be detected. The authors should clarify why they are unable to present the data on ILC3s, neutrophils and macrophage in sputum.

RESPONSE: First, let us talk about the cellularity of ILCs in the sputum. The total number of cells collected from sputum averaged 3.5×10^4 cells per ml of sputum. Among them, the average frequency of CD45+ cells was about 15%, and that of lymphocytes was only 4.5% of CD45+ cells (This value is consistent with the results of reference, Spanevello A et al. Am J Respir Crit Care Med. 2000). Furthermore, ILCs make up less than 1% of lymphocytes. Therefore, the ILC is about 25 per ml of sputum, and the absolute number of ILCs we can obtain is 200-300. Due to the rarity of ILC, we had to devote most of the sputum to the ILC analysis. Measuring cytokines

after stimulation in such a sparse population is not a good idea. Because we think it is hard to get reliable results.

Second, this study aimed to investigate whether cigarette smoke alters ILCs and the effect of ILCs on lung function in asthmatics. Therefore, we did not analyze the frequency of neutrophils, as we focused on ILC and smoking-related changes over other cells in the sputum. Instead, we analyzed macrophages because there was a correlation between ILCs and macrophages in our previous study (Kim et al, J Allergy Clin Immunol 2019), and macrophages are the major cells of the sputum. Although the properties of macrophages in sputum and blood are significantly different, neutrophils do not (Lavinskiene S et al. Inflamm Res. 2014).

[REDACTED]

Please understand that we have no choice but to focus on our desired goal with limited specimens.

Reviewer #3 (Remarks to the Author):

COMMENT: The authors responded to my criticism and revised the manuscript accordingly. The revised manuscript is acceptable for publication.

RESPONSE: We greatly thank reviewer #3 for recommending our manuscript is acceptable for publication.